# Influencing Long-Term Behavior in Multiagent Reinforcement Learning

**Dong-Ki Kim**[1,3]
dkkim93@mit.edu

**Matthew Riemer**[2,3,4]
mdriemer@us.ibm.com

**Miao Liu**[2,3]
miao.liu1@us.ibm.com

**Jakob N. Foerster**[5]
jakob.foerster@eng.ox.ac.uk

**Michael Everett**[1,3]
mfe@mit.edu

**Chuangchuang Sun**[1,3]
ccsun1@mit.edu

**Gerald Tesauro**[2,3]
gtesauro@us.ibm.com

**Jonathan P. How**[1,3]
jhow@mit.edu

## Abstract

The main challenge of multiagent reinforcement learning is the difficulty of learning useful policies in the presence of other simultaneously learning agents whose changing behaviors jointly affect the environment's transition and reward dynamics. An effective approach that has recently emerged for addressing this non-stationarity is for each agent to anticipate the learning of other agents and influence the evolution of future policies towards desirable behavior for its own benefit. Unfortunately, previous approaches for achieving this suffer from myopic evaluation, considering only a finite number of policy updates. As such, these methods can only influence transient future policies rather than achieving the promise of scalable equilibrium selection approaches that influence the behavior at convergence. In this paper, we propose a principled framework for considering the limiting policies of other agents as time approaches infinity. Specifically, we develop a new optimization objective that maximizes each agent's average reward by directly accounting for the impact of its behavior on the limiting set of policies that other agents will converge to. Our paper characterizes desirable solution concepts within this problem setting and provides practical approaches for optimizing over possible outcomes. As a result of our farsighted objective, we demonstrate better long-term performance than state-of-the-art baselines across a suite of diverse multiagent benchmark domains.

## 1 Introduction

Learning in multiagent reinforcement learning (MARL) is fundamentally difficult because an agent interacts with other simultaneously learning agents in a shared environment [1]. The joint learning of agents induces non-stationary environment dynamics from the perspective of each agent, requiring an agent to adapt its behavior with respect to potentially unknown changes in the policies of other agents [2]. Notably, non-stationary policies will converge to a recurrent set of joint policies by the end of learning. In practice, this converged joint policy can correspond to a game-theoretic solution concept, such as a Nash equilibrium [3] or more generally a cyclic correlated equilibrium [4], but multiple equilibria can exist for a single game with some of these Pareto dominating others [5]. Hence, a critical question in addressing this non-stationarity is how individual agents should behave to influence convergence of the recurrent set of policies towards more desirable limiting behaviors.

Our key idea in this work is to consider the limiting policies of other agents as time approaches infinity. Specifically, the converged behavior of this dynamic multiagent system is not due to some

---

[1]MIT-LIDS  [2]IBM-Research  [3]MIT-IBM Watson AI Lab  [4]Mila  [5]University of Oxford

36th Conference on Neural Information Processing Systems (NeurIPS 2022).

arbitrary stochastic processes, but rather each agent's underlying learning process that also depends on the behaviors of the other interacting agents. As such, effective agents should model how their actions can influence the limiting behavior of other agents and leverage those dependencies to shape the convergence process. This farsighted perspective contrasts with recent work that also considers influencing the learning of other agents [6–11]. While these approaches show improved performance over methods that neglect the learning of other agents entirely [12–14], they suffer from myopic evaluation: only considering a few updates to the policies of other agents or optimizing for the discounted return, which only considers a finite horizon time of $1/(1-\gamma)$ for discount factor $\gamma$ [15].

**Our contribution.** With this insight, we make the following primary contributions in this paper:

• **Formalization of multiagent non-stationarity (Section 2).** We introduce an active Markov game setting that formalizes MARL with simultaneously learning agents as a directed graphical model and captures the underlying non-stationarity over time. We detail how such a system eventually converges to a stationary periodic distribution. As such, the objective is to maximize its long-term rewards over this distribution and, if each agent maximizes this objective, the resulting multiagent system settles into a new and general equilibrium concept that we call an active equilibrium.

• **Practical framework for optimizing an active Markov game (Section 3).** We outline a practical approach for optimization in this setting, called FUlly Reinforcing acTive influence witH averagE Reward (FURTHER). Our approach is based on a policy gradient and Bellman update rule tailored to active Markov games. Moreover, we show how variational inference can be used to approximate the update function of other agents and support decentralized execution and training.

• **Comprehensive evaluation of our approach (Section 4).** We demonstrate that our method consistently converges to a more desirable limiting distribution than baseline methods that either neglect the learning of others [14] or consider their learning with a myopic perspective [8] in various multiagent benchmark domains. We also demonstrate that FURTHER provides a flexible framework such that it can incorporate recent advances in multiagent learning and improve performance in large-scale settings by leveraging the mean-field method [16].

## 2 Problem Statement: Active Markov Game

This work studies a general multiagent learning setting, where each agent interacts with other independently learning agents in a shared environment. Agents in this setting update their policies based on recent experiences which are affected by the joint actions of all agents. As such, while an agent cannot directly modify the future policies of other interacting agents, the agent can actively influence them by changing its own actions. In this section, we first formalize the presence of this causal influence in multiagent interactions by introducing the new framework of an active Markov game. We then formalize solution concepts and objectives for learning within this framework. Finally, we discuss dependence on initial states and policies, detailing choices that we can make to minimize the impact of these initial conditions on behavior after convergence.

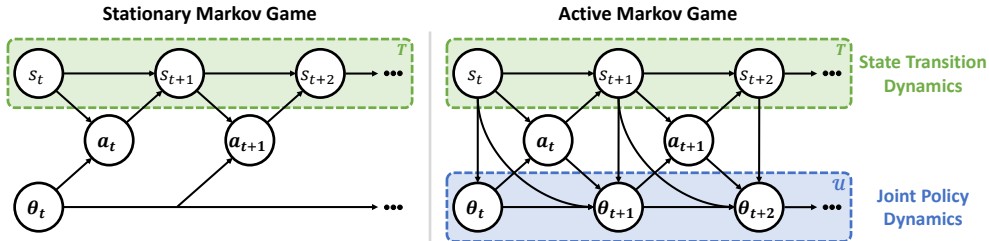

Figure 1: Within the stationary Markov game setting, agents wrongly assume that other agents will have stationary policies into the future. In contrast, agents in an active Markov game recognize that other agents have non-stationary policies based on the Markovian update functions.

### 2.1 Directed Graphical Model of Active Markov Game

We define an active Markov game as a tuple $\mathcal{M}_n = \langle \mathcal{I}, \mathcal{S}, \mathcal{A}, \mathcal{T}, \mathcal{R}, \Theta, \mathcal{U} \rangle$; $\mathcal{I} = \{1, \ldots, n\}$ is the set of $n$ agents; $\mathcal{S}$ is the state space; $\mathcal{A} = \times_{i \in \mathcal{I}} \mathcal{A}^i$ is the joint action space; $\mathcal{T} : \mathcal{S} \times \mathcal{A} \mapsto \mathcal{S}$ is the state transition function; $\mathcal{R} = \times_{i \in \mathcal{I}} \mathcal{R}^i$ is the joint reward function; $\Theta = \times_{i \in \mathcal{I}} \Theta^i$ is the joint policy

parameter space; and $\mathcal{U} = \times_{i \in \mathcal{I}} \mathcal{U}^i$ is the joint Markovian policy update function. We typeset sets in bold for clarity. Compared to the stationary Markov game that effectively represents MARL with wrongly assumed stationary policies in the future, the active Markov game considers how policies change over time (see Figure 1). Specifically, at each timestep $t$, each agent $i$ executes an action at a current state $s_t \in \mathcal{S}$ according to its stochastic policy $a_t^i \sim \pi^i(\cdot|s_t; \theta_t^i)$ parameterized by $\theta_t^i \in \Theta^i$. A joint action $\boldsymbol{a_t} = \{a_t^i, \boldsymbol{a_t^{-i}}\}$ yields a transition from $s_t$ to $s_{t+1}$ with probability $\mathcal{T}(s_{t+1}|s_t, \boldsymbol{a_t})$, where the notation $\boldsymbol{-i}$ indicates all other agents except agent $i$. Each agent $i$ then obtains a reward according to its reward function $r_t^i = \mathcal{R}^i(s_t, \boldsymbol{a_t})$ and updates its policy parameters according to $\mathcal{U}^i(\theta_{t+1}^i|\theta_t^i, \tau_t^i)$, where $\tau_t^i = \{s_t, \boldsymbol{a_t}, r_t^i, s_{t+1}\}$ denotes agent $i$'s transition. This process continues until the convergence of non-stationary policies. Notably, the joint policy update function $\mathcal{U}$ is a function of $a_t^i$, which affects the state transitions and rewards, so agent $i$ can actively influence future joint policies by changing its own behavior. Modeling this influence rather than ignoring it is the main advantage of using active Markov games rather than the stationary Markov game formalism.

## 2.2 Solution Concepts in Active Markov Games

The formalism of active Markov games provides a principled framework for each agent to model the impact of its behavior on joint future policies. In this section, we study the theoretical convergence properties of an active Markov game and develop relevant terminology that will help us characterize this convergence. We begin by formalizing the limiting behavior as a stationary periodic distribution.

**Definition 1.** (Stationary $k$-Periodic Distribution). *The limiting behavior of an active Markov game can be represented by a stationary periodic probability distribution over the joint space of states and policies, defined as a stationary conditional distribution with respect to a period of order $k$:*

$$\mu_k(s, \boldsymbol{\theta}|s_0, \boldsymbol{\theta_0}, \ell) = p(s_t = s, \boldsymbol{\theta_t} = \boldsymbol{\theta}|s_0, \boldsymbol{\theta_0}, \ell) \quad \forall t \geq 0, s, s_0 \in \mathcal{S}, \boldsymbol{\theta}, \boldsymbol{\theta_0} \in \boldsymbol{\Theta}, \quad (1)$$

*where $\ell = t\%k$ with % denoting the modulo operation. The stationary $k$-periodic distribution satisfies the following property as its time averaged expectation stays stationary in the limit:*

$$\frac{1}{k}\sum_{\ell=1}^{k}\mu_k(s_{\ell+1}, \boldsymbol{\theta_{\ell+1}}|s_0, \boldsymbol{\theta_0}, \ell+1) = \frac{1}{k}\sum_{\ell=1}^{k}\sum_{s_\ell, \boldsymbol{\theta_\ell}}\mu_k(s_\ell, \boldsymbol{\theta_\ell}|s_0, \boldsymbol{\theta_0}, \ell)\sum_{\boldsymbol{a_\ell}}\boldsymbol{\pi}(\boldsymbol{a_\ell}|s_\ell; \boldsymbol{\theta_\ell})$$
$$\mathcal{T}(s_{\ell+1}|s_\ell, \boldsymbol{a_\ell})\,\mathcal{U}(\boldsymbol{\theta_{\ell+1}}|\boldsymbol{\theta_\ell}, \boldsymbol{\tau_\ell}) \quad \forall s_{\ell+1} \in \mathcal{S}, \boldsymbol{\theta_{\ell+1}} \in \boldsymbol{\Theta}. \quad (2)$$

Our notion of a stationary $k$-periodic distribution provides a flexible representation for characterizing the limiting distribution, generalizing from fully stationary fixed-point convergence (when $k=1$) to the extreme case of totally non-stationary convergence (when $k \to \infty$).

Having defined the joint convergence behavior of an active Markov game, we can now develop an objective that each agent can optimize to maximize its long-term rewards. Our key finding is that the average reward formulation, developed for single-agent learning [17, 18], is well suited for studying the limiting behavior of other interacting agents in multiagent learning. In particular, the average reward formulation maximizes the agent's average reward per step with equal weight given to immediate and delayed rewards, unlike the discounted return objective. Once the joint policy converges to the stationary periodic distribution, rewards collected by this recurrent set of policies govern each agent's average reward as $t \to \infty$. Thus, optimizing for the average reward in an active Markov game encourages agents to consider how to influence the limiting set of policies after convergence rather than transient policies that are only experienced momentarily.

**Definition 2.** (Active Average Reward Objective). *Each agent $i$ in an active Markov game aims to find policy parameters $\theta^i$ and update function $\mathcal{U}^i$ that maximize its expected average reward $\rho^i \in \mathbb{R}$:*

$$\max_{\theta^i, \mathcal{U}^i} \rho^i(s, \boldsymbol{\theta}, \mathcal{U}) := \max_{\theta^i, \mathcal{U}^i} \lim_{T \to \infty} \mathbb{E}\Big[\frac{1}{T}\sum_{t=0}^{T}\mathcal{R}^i(s_t, \boldsymbol{a_t})\Big|^{\substack{s_0=s, \boldsymbol{\theta_0}=\boldsymbol{\theta}, \\ \boldsymbol{a_{0:T}} \sim \boldsymbol{\pi}(\cdot|s_{0:T}; \boldsymbol{\theta_{0:T}}), \\ s_{t+1} \sim \mathcal{T}(\cdot|s_t, \boldsymbol{a_t}), \boldsymbol{\theta_{t+1}} \sim \mathcal{U}(\cdot|\boldsymbol{\theta_t}, \boldsymbol{\tau_t})}}\Big]$$
$$= \max_{\theta^i, \mathcal{U}^i} \frac{1}{k}\sum_{\ell=1}^{k}\sum_{s_\ell, \boldsymbol{\theta_\ell}}\mu_k(s_\ell, \boldsymbol{\theta_\ell}|s, \boldsymbol{\theta}, \ell)\sum_{\boldsymbol{a_\ell}}\boldsymbol{\pi}(\boldsymbol{a_\ell}|s_\ell; \boldsymbol{\theta_\ell})\mathcal{R}^i(s_\ell, \boldsymbol{a_\ell}), \quad (3)$$

where $T$ denotes the time horizon. It is important to note that Equation (3) has no preference over the large equivalence class of update functions that eventually converge to an optimal limiting behavior, and we only require finding an update function in this class even if the convergence rate is slow. This is advantageous for our discussion to come about solution concepts in active Markov games. However,

in our practical approach to optimization, we also optimize over the transient distribution, pushing towards solutions with lower regret by modeling our value function based on the differential returns as in Proposition 2. We also note that even if a single agent maximizes this objective, agents will not necessarily arrive at any kind of equilibrium. This is because other agents may have sub-optimal or biased update functions beyond the agent's control, and a rational agent can potentially seek to converge to an average reward that is better for it than that of any equilibrium as a result. Additionally, whether an agent just seeks to optimize its policy or maximize its update function as well depends on the kind of solution concept that is desired. For example, finding a fixed stationary policy equates to using an update function that arrives at a fixed point, whereas we can also optimize over the update function if we seek to find an optimal non-stationary policy as in the meta-learning literature [9, 19].

If all agents maximize the active average reward objective, we arrive at a new and general equilibrium concept that we call an active equilibrium, where no agents can further optimize its average reward:

**Definition 3.** (Active Equilibrium). *In an active Markov game, an active equilibrium is joint policy parameters* $\boldsymbol{\theta}^* = \{\theta^{i*}, \boldsymbol{\theta}^{-i*}\}$ *with associated joint update function* $\boldsymbol{\mathcal{U}}^* = \{\mathcal{U}^{i*}, \boldsymbol{\mathcal{U}}^{-i*}\}$ *such that*:

$$\rho^i(s, \theta^{i*}, \boldsymbol{\theta}^{-i*}, \mathcal{U}^{i*}, \boldsymbol{\mathcal{U}}^{-i*}) \geq \rho^i(s, \theta^i, \boldsymbol{\theta}^{-i*}, \mathcal{U}^i, \boldsymbol{\mathcal{U}}^{-i*}) \quad \forall i \in \mathcal{I}, s \in \mathcal{S}, \theta^i \in \Theta^i, \mathcal{U}^i \in \mathbb{U}^i. \quad (4)$$

where $\mathbb{U}^i$ denotes the space of agent $i$'s update functions. Our active equilibrium definition is related to non-stationary solution concepts in game theory, such as the non-stationary Nash equilibrium [20], that search for a sequence of best-response joint policies. However, these non-stationary solutions are generally intractable to compute due to the unconstrained sequence over the infinite horizon and the resulting large policy search space size. By contrast, the active equilibrium provides a more refined and practical notion than these solution concepts by having a constraint on the sequence based on the update functions. We also note the generality of active equilibrium that it can correspond to other standard solution concepts as we impose restrictions on relevant variables:

**Remark 1.** (Connection to Existing Solution Concepts). *Stationary Nash [3] and correlated equilibria [21] are special kinds of active equilibria when* $k = 1$ *and joint action distributions are independent and correlated, respectively. Cyclic Nash and cyclic correlated equilibria [4] are also special cases of an active equilibrium if* $k > 1$*, the joint update function is deterministic, and joint action distributions are independent and correlated, respectively.*

## 2.3 Addressing Sensitivity to Initial Conditions

The recurrent set of converged joint policies is generally dependent on initial states and policies, as specified by the conditioned initial variables in Equations (1) to (3). This initial condition dependence implies that there can be instances of poor convergence performance simply due to undesirable initial states and policies (see Appendix A for an example). In this paper, we address this sensitivity to initial conditions by considering the stochastically stable periodic distribution, which is a special case of the stationary periodic distribution. The stochastic distribution describes the limiting joint behavior when each agent has communicating strategies (i.e., for every pair of policy parameters $\theta^i, \theta^{i\prime} \in \Theta^i$, $\theta^i$ transitions to $\theta^{i\prime}$ in a finite number of steps with non-zero probability and vice versa) by adding noise $\epsilon$ to its update function $\mathcal{U}^i_\epsilon$, and noise $\epsilon \to 0$ over time (i.e., $\lim_{t\to\infty} \mathcal{U}^i_\epsilon = \mathcal{U}^i$). Importantly, the stochastic distribution provides an important analytical benefit of independent convergence with respect to the initial conditions. Specifically, assuming communicating state transitions $\mathcal{T}$, if only agent $i$'s update function is perturbed, then we arrive at the notion of self-stable periodic distribution:

**Definition 4.** (Self-Stable Periodic Distribution). *Given communicating state transition* $\mathcal{T}$*, if noise* $\epsilon$ *is added only to the agent* $i$*'s update function* $\mathcal{U}^i_\epsilon$*, we achieve the stationary* $k$*-periodic distribution independent of the initial state and the agent* $i$*'s initial policy as* $\epsilon \to 0$ *over time*:

$$\frac{1}{k}\sum_{\ell=1}^{k}\mu_k(s_{\ell+1}, \boldsymbol{\theta_{\ell+1}}|\boldsymbol{\theta_0^{-i}}, \ell+1) = \frac{1}{k}\sum_{\ell=1}^{k}\sum_{s_\ell, \boldsymbol{\theta_\ell}}\mu_k(s_\ell, \boldsymbol{\theta_\ell}|\boldsymbol{\theta_0^{-i}}, \ell)\sum_{\boldsymbol{a_\ell}}\boldsymbol{\pi}(\boldsymbol{a_\ell}|s_\ell; \boldsymbol{\theta_\ell})$$
$$\mathcal{T}(s_{\ell+1}|s_\ell, \boldsymbol{a_\ell})\,\boldsymbol{\mathcal{U}}(\boldsymbol{\theta_{\ell+1}}|\boldsymbol{\theta_\ell}, \boldsymbol{\tau_\ell}) \quad \forall s_{\ell+1} \in \mathcal{S}, \boldsymbol{\theta_{\ell+1}} \in \boldsymbol{\Theta}. \quad (5)$$

Similarly, if the full joint update function is perturbed with noise $\boldsymbol{\mathcal{U}}_\epsilon$, this induces a unique stationary periodic distribution independent of the initial state and initial joint policy:

**Definition 5.** (Jointly-Stable Periodic Distribution). *Given communicating state transition* $\mathcal{T}$*, if noise* $\epsilon$ *is added to the joint update function* $\boldsymbol{\mathcal{U}}_\epsilon$*, we achieve the same stationary* $k$*-periodic distribution*

*independent of the initial state and the initial policies as $\epsilon \to 0$ over time*:

$$\frac{1}{k}\sum_{\ell=1}^{k}\mu_k(s_{\ell+1}, \boldsymbol{\theta_{\ell+1}}|\ell+1) = \frac{1}{k}\sum_{\ell=1}^{k}\sum_{s_\ell, \boldsymbol{\theta_\ell}}\mu_k(s_\ell, \boldsymbol{\theta_\ell}|\ell)\sum_{\boldsymbol{a_\ell}}\boldsymbol{\pi}(\boldsymbol{a_\ell}|s_\ell; \boldsymbol{\theta_\ell}) \tag{6}$$
$$\mathcal{T}(s_{\ell+1}|s_\ell, \boldsymbol{a_\ell})\,\mathcal{U}(\boldsymbol{\theta_{\ell+1}}|\boldsymbol{\theta_\ell}, \boldsymbol{\tau_\ell}) \quad \forall s_{\ell+1} \in \mathcal{S}, \boldsymbol{\theta_{\ell+1}} \in \boldsymbol{\Theta}.$$

**Proposition 1.** (Uniqueness of Jointly-Stable Periodic Distribution). *Given communicating state transition $\mathcal{T}$ and perturbed joint update function with noise $\mathcal{U}_\epsilon$, the jointly-stable periodic distribution is unique as $\epsilon \to 0$ over time.*

*Proof.* See Appendix B for details. □

The jointly-stable periodic distribution is induced in many cases of interest to multiagent learning, including when all policies employ update functions leveraging the Greedy in the Limit with Infinite Exploration (GLIE) property [18]: 1) all state-action pairs are visited infinitely often and 2) as $t \to \infty$, the behavior policy converges to the greedy policy.

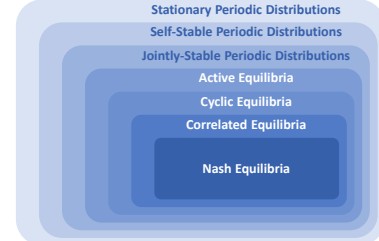

In particular, a broad class of action exploration or noisy policy update functions lead to this kind of distribution [22–26]. Indeed, MARL algorithms generally rely on persistent exploration and thus satisfy GLIE. Lastly, as demonstrated in Figure 2, although maximizing over the space of jointly-stable periodic distributions confines the search space of stationary periodic distributions, the best possible active equilibria still lie within this smaller space while also allowing for optimization robust to initial conditions. We focus on designing a learning algorithm that can find an equilibrium in the practical and confined search space of the jointly-stable distributions in the following section.

Figure 2: Venn diagram describing relationships between the proposed distributions and equilibrium concepts.

## 3 FURTHER: Practical Method for Solving Active Markov Game

In this section, we develop a practical method, called FURTHER, for learning beneficial policies in the space of the jointly-stable periodic distributions. We first outline a practical version of the average reward objective and derive its policy gradient. We then detail our model-free implementation that builds on top of soft actor-critic [27] and variational inference [28] to learn policies that efficiently optimize for the average reward objective in a decentralized manner.

### 3.1 Formulation and Policy Gradient Theorem of FURTHER

While the objective in Equation (3) ideally maximizes over the space of update functions and learns a non-stationary policy, addressing the computational difficulty of long horizon meta-learning still remains an active area of research [9, 29, 30]. As such, in FURTHER, we take a practical step forward and learn the optimal fixed point policy that influences joint policy behavior to maximize its long-term average reward $\rho_{\theta^i}^i \in \mathbb{R}$ at a state $s \in \mathcal{S}$ and policy parameters of other agents $\boldsymbol{\theta^{-i}} \in \boldsymbol{\Theta^{-i}}$:

$$\max_{\theta^i} \rho_{\theta^i}^i(s, \boldsymbol{\theta^{-i}}) := \max_{\theta^i}\lim_{T \to \infty}\mathbb{E}\left[\frac{1}{T}\sum_{t=0}^{T}\mathcal{R}^i(s_t, \boldsymbol{a_t})\,\Bigg|\,\substack{s_0=s,\ \boldsymbol{\theta_0^{-i}}=\boldsymbol{\theta^{-i}},\\ a_{0:T}^i \sim \pi(\cdot|s_{0:T};\theta^i), \boldsymbol{a_{0:T}^{-i}} \sim \boldsymbol{\pi}(\cdot|s_{0:T};\boldsymbol{\theta_{0:T}^{-i}}),\\ s_{t+1} \sim \mathcal{T}(\cdot|s_t, \boldsymbol{a_t}), \boldsymbol{\theta_{t+1}^{-i}} \sim \boldsymbol{\mathcal{U}^{-i}}(\cdot|\boldsymbol{\theta_t^{-i}}, \boldsymbol{\tau_t^{-i}})}\right], \tag{7}$$

where the subscript $\theta^i$ notation denotes the implicit dependence on the learning of agent $i$'s fixed stationary policy. As discussed in Section 2.3, a useful result under the jointly-stable periodic distribution is that the average reward becomes independent of the initial states and policies:

$$\rho_{\theta^i}^i(s, \boldsymbol{\theta^{-i}}) = \rho_{\theta^i}^i(s', \boldsymbol{\theta^{-i\prime}}) = \rho_{\theta^i}^i \quad \forall s \neq s', \boldsymbol{\theta^{-i}} \neq \boldsymbol{\theta^{-i\prime}}. \tag{8}$$

We now derive the Bellman equation in an active Markov game that defines the relationship between the value function and average reward.

**Proposition 2.** (Active Differential Bellman Equation). *The differential value function $v_{\theta^i}^i$ represents the expected total difference between the accumulated rewards from $s$ and $\boldsymbol{\theta^{-i}}$ and the average reward*

$\rho_{\theta^i}^i$ [18]. *The differential value function inherently includes the recursive relationship with respect to $v_{\theta^i}^i$ at the next state $s'$ and the updated policies of other agents $\boldsymbol{\theta}^{-i\prime}$:*

$$v_{\theta^i}^i(s, \boldsymbol{\theta^{-i}}) = \lim_{T\to\infty} \mathbb{E}\Big[\sum_{t=0}^{T}\big(\mathcal{R}^i(s_t, \boldsymbol{a_t}) - \rho_{\theta^i}^i\big)\Big| \begin{smallmatrix} s_0=s,\,\boldsymbol{\theta}_0^{-i}=\boldsymbol{\theta}^{-i},\\ a_{0:T}^i\sim\pi(\cdot|s_{0:T};\theta^i),\boldsymbol{a}_{0:T}^{-i}\sim\boldsymbol{\pi}(\cdot|s_{0:T};\boldsymbol{\theta}_{0:T}^{-i}),\\ s_{t+1}\sim\mathcal{T}(\cdot|s_t,\boldsymbol{a_t}),\boldsymbol{\theta}_{t+1}^{-i}\sim\boldsymbol{\mathcal{U}}^{-i}(\cdot|\boldsymbol{\theta}_t^{-i},\boldsymbol{\tau}_t^{-i}) \end{smallmatrix}\Big]$$

$$= \sum_{a^i}\pi(a^i|s;\theta^i)\sum_{\boldsymbol{a^{-i}}}\boldsymbol{\pi}(\boldsymbol{a^{-i}}|s;\boldsymbol{\theta^{-i}})\sum_{s'}\mathcal{T}(s'|s,\boldsymbol{a})\sum_{\boldsymbol{\theta}^{-i\prime}}\boldsymbol{\mathcal{U}^{-i}}(\boldsymbol{\theta}^{-i\prime}|\boldsymbol{\theta}^{-i},\boldsymbol{\tau}^{-i}) \qquad (9)$$
$$\Big[\mathcal{R}^i(s,\boldsymbol{a}) - \rho_{\theta^i}^i + v_{\theta^i}^i(s', \boldsymbol{\theta}^{-i\prime})\Big].$$

*Proof.* See Appendix C for a derivation. $\qquad\qquad\square$

Finally, we derive the policy gradient based on the active differential Bellman equation:

**Proposition 3.** (Active Average Reward Policy Gradient Theorem). *The gradient of active average reward objective in Equation (7) with respect to agent $i$'s policy parameters $\theta^i$ is:*

$$\nabla_{\theta^i}J_\pi^i(\theta^i) = \frac{1}{k}\sum_{\ell=1}^{k}\sum_{s_\ell,\boldsymbol{\theta}_\ell^{-i}}\mu_{k,\theta^i}(s_\ell, \boldsymbol{\theta_\ell}|\ell)\sum_{a_\ell^i}\nabla_{\theta^i}\pi(a_\ell^i|s_\ell;\theta^i)\sum_{\boldsymbol{a_\ell^{-i}}}\boldsymbol{\pi}(\boldsymbol{a_\ell^{-i}}|s_\ell;\boldsymbol{\theta_\ell^{-i}})q_{\theta^i}^i(s_\ell, \boldsymbol{\theta_\ell^{-i}}, \boldsymbol{a_\ell}), \quad (10)$$

*with* $q_{\theta^i}^i(s_\ell, \boldsymbol{\theta_\ell^{-i}}, \boldsymbol{a_\ell}) = \sum_{s_{\ell+1}}\mathcal{T}(s_{\ell+1}|s_\ell, \boldsymbol{a_\ell})\sum_{\boldsymbol{\theta_{\ell+1}^{-i}}}\boldsymbol{\mathcal{U}^{-i}}(\boldsymbol{\theta_{\ell+1}^{-i}}|\boldsymbol{\theta_\ell^{-i}}, \boldsymbol{\tau_\ell^{-i}})\Big[\mathcal{R}^i(s_\ell, \boldsymbol{a_\ell}) - \rho_{\theta^i}^i + v_{\theta^i}^i(s_{\ell+1}, \boldsymbol{\theta_{\ell+1}^{-i}})\Big].$

*Proof.* See Appendix D for a derivation. $\qquad\qquad\square$

### 3.2 Soft Actor-Critic Implementation with Variational Inference

**Algorithm overview.** FURTHER broadly consists of inference and reinforcement learning modules. In practice, each agent has partial observations about others and cannot directly observe their true policy parameters $\boldsymbol{\theta^{-i}}$ and policy dynamics $\boldsymbol{\mathcal{U}^{-i}}$. The inference learning module predicts this hidden information about other agents leveraging variational inference [28] modified for sequential prediction. The inferred information becomes the input to the reinforcement learning module, which extends the policy gradient theorem in Equation (10) and learns active average reward policies sample efficiently by building on the multiagent soft actor-critic (MASAC) framework [14, 27, 31]. We note that each agent interacts and learns these modules by only observing the actions of other agents, so our implementation supports decentralized execution and training. We provide further details, including implementation for $k>1$ and psuedocode, in Appendix E.

For simplicity, we consider the period $k=1$ and develop corresponding soft reinforcement learning optimizations in Equations (12) to (14).

**Inference learning module.** This module aims to infer the current policies of other agents and their learning dynamics based on an approximate variational inference [28]. Specifically, we optimise a tractable evidence lower bound (ELBO), defined together with an encoder $p(\hat{\boldsymbol{z}}_{t+1}^{-i}|\hat{\boldsymbol{z}}_t^{-i}, \tau_t^i; \phi_{\text{enc}}^i)$ and a decoder $p(\boldsymbol{a_t^{-i}}|s_t, \hat{\boldsymbol{z}}_t^{-i}; \phi_{\text{dec}}^i)$, parameterised by $\phi_{\text{enc}}^i$ and $\phi_{\text{dec}}^i$, respectively:

$$\mathcal{J}_{\text{elbo}}^i = \mathbb{E}_{p(\tau_{0:t}),p(\hat{\boldsymbol{z}}_{1:t}^{-i}|\tau_{0:t-1};\phi_{\text{enc}}^i)}\Big[\sum_{t'=1}^{t}\underbrace{\log p(\boldsymbol{a_{t'}^{-i}}|s_{t'}, \hat{\boldsymbol{z}}_{t'}^{-i}; \phi_{\text{dec}}^i)}_{\text{Reconstruction loss}} - \underbrace{D_{\text{KL}}\big(p(\hat{\boldsymbol{z}}_{t'}^{-i}|\tau_{t'-1}^i;\phi_{\text{enc}}^i)||p(\hat{\boldsymbol{z}}_{t'-1}^{-i})\big)}_{\text{KL divergence}}\Big], \quad (11)$$

where latent strategies $\hat{\boldsymbol{z}}_t^{-i}$ represents inferred policy parameters of other agents $\boldsymbol{\theta}_t^{-i}$, the encoder represents the policy dynamics of other agents $\boldsymbol{\mathcal{U}^{-i}}$ with parameters $\phi_{\text{enc}}^i$, and $\tau_{0:t}^i = \{\tau_0^i, ..., \tau_t^i\}$ denotes $i$'s transitions up to timestep $t$. We refer to Appendix F for a detailed ELBO derivation. By optimizing the reconstruction term, the encoder aims to infer accurate next latent strategies of other agents. Also, by imposing the prior through the KL divergence, where we set the prior to the previous posterior with initial prior $p(\hat{\boldsymbol{z}}_0^{-i}) = \mathcal{N}(0, I)$, the inferred policies from the encoder are encouraged to be sequentially consistent across time (i.e., no abrupt changes in policies of others).

**Reinforcement learning module.** This module aims to learn a policy that can maximize the agent's average reward based on the inferred information about other agents. Each agent maintains its policy $\pi(\cdot|s, \hat{\boldsymbol{z}}^{-i}; \theta^i)$ parameterized by $\theta^i$, two $q$-functions $q_{\theta^i}^i(s, \hat{\boldsymbol{z}}^{-i}, \boldsymbol{a}; \psi_1^i)$ and $q_{\theta^i}^i(s, \hat{\boldsymbol{z}}^{-i}, \boldsymbol{a}; \psi_2^i)$

|   | B | S |
|---|---|---|
| B | (2, 1) | (0, 0) |
| S | (0, 0) | (1, 2) |

(a) Bach/Stravinsky

|   | U | D |
|---|---|---|
| U | (4, 4) | (0, 0) |
| D | (0, 0) | (8, 8) |

(b) Coordination

|   | H | T |
|---|---|---|
| H | (1, -1) | (-1, 1) |
| T | (-1, 1) | (1, -1) |

(c) Matching Pennies

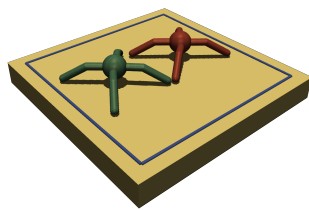

(d) MuJoCo RoboSumo

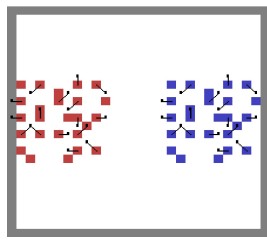

(e) MAgent Battle

Figure 3: **(a)-(c)** Payoff tables for Bach or Stravinsky (general-sum), coordination (cooperative), and matching pennies (competitive) games. **(d)** A competitive RoboSumo domain [19] with two agents fighting each other. **(e)** A mixed cooperative-competitive battle domain [32] with 25 vs 25 agents.

parameterized by $\psi_1^i, \psi_2^i$, and learnable average reward $\rho_{\theta^i}^i \in \mathbb{R}$. We train the q-functions and $\rho_{\theta^i}^i$ by minimizing the soft Bellman residual:

$$J_q^i(\psi_\beta^i, \rho_{\theta^i}^i) = \mathbb{E}_{(s, \hat{\boldsymbol{z}}^{-i}, \boldsymbol{a}, r^i, s', \hat{\boldsymbol{z}}^{-i\prime}) \sim \mathcal{D}^i}\Big[\big(y - q_{\theta^i}^i(s, \hat{\boldsymbol{z}}^{-i}, \boldsymbol{a}; \psi_\beta^i)\big)^2\Big], \ \ y = r^i - \rho_{\theta^i}^i + v_{\theta^i}^i(s', \hat{\boldsymbol{z}}^{-i\prime}; \bar{\psi}_\beta^i), \quad (12)$$

where $\beta = 1, 2$, $\mathcal{D}^i$ denotes $i$'s replay buffer, and $\bar{\psi}_\beta^i$ denotes the target q-network parameters. The soft value function $v_{\theta^i}^i$ calculates the state value with the policy entropy $\mathcal{H}$ and entropy weight $\alpha$:

$$v_{\theta^i}^i(s, \hat{\boldsymbol{z}}^{-i}; \psi^i) = \sum_{a^i} \pi(a^i | s, \hat{\boldsymbol{z}}^{-i}; \theta^i) \sum_{\boldsymbol{a}^{-i}} \pi(\boldsymbol{a}^{-i} | s; \hat{\boldsymbol{z}}^{-i}) \min_{\beta=1,2} q_{\theta^i}^i(s, \hat{\boldsymbol{z}}^{-i}, \boldsymbol{a}; \psi_\beta^i) + \alpha \mathcal{H}(\pi(\cdot | s, \hat{\boldsymbol{z}}^{-i}; \theta^i)). \quad (13)$$

Finally, the policy is trained to maximize:

$$J_\pi^i(\theta^i) = \mathbb{E}_{(s, \hat{\boldsymbol{z}}^{-i}, \boldsymbol{a}^{-i}) \sim \mathcal{D}^i}\Big[\sum_{a^i} \pi(a^i | s, \hat{\boldsymbol{z}}^{-i}; \theta^i) \min_{\beta=1,2} q_{\theta^i}^i(s, \hat{\boldsymbol{z}}^{-i}, \boldsymbol{a}; \psi_\beta^i) + \alpha \mathcal{H}(\pi(\cdot | s, \hat{\boldsymbol{z}}^{-i}; \theta^i))\Big]. \quad (14)$$

We note that Equations (13) and (14) are for discrete action space, and we detail optimizations for continuous action space in Appendix E.

**Mean-Field FURTHER.**    FURTHER provides a flexible framework such that it can easily integrate recent advances in multiagent learning. For example, by reconstructing and predicting the mean action and latent strategy of neighbor agents in Equation (11), we can incorporate the mean-field framework to improve performance in large-scale learning settings. Appendix E details the mean-field version of FURTHER with pseudocode.

## 4 Evaluation

We demonstrate FURTHER's efficacy on a diverse suite of multiagent benchmark domains. We refer to Appendix G for experimental details and hyperparameters. The code is available at `https://bit.ly/3fXArAo`, and video highlights are available at `https://bit.ly/37IWeb9`. The mean and 95% confidence interval computed for 20 seeds are shown in each figure.

**Baselines.**    We compare FURTHER with the following baselines (see Appendix G.2 for details):

• **LILI [8]:** An approach that considers the learning dynamics of other agents but suffers from myopic evaluation bias by optimizing the discounted return objective (see Equation (25)).

• **MASAC [14]:** An approach that extends SAC [27] to a multiagent learning setting by having centralized critics [12]. This baseline assumes other agents will have stationary policies in the future and thus neglects their learning (see Equation (26)).

We note that these selected baselines are closely related to FURTHER, optimizing different objectives with respect to $\mathcal{U}^{-i}$. In particular, LILI and MASAC optimize the discounted return objective with and without modeling $\mathcal{U}^{-i}$, respectively. As such, our baseline choices enable us to separately analyze the effect of FURTHER's novel average reward objective. For completeness, we also consider additional baselines of an opponent modeling method (DRON) and an incentive MARL method (MOA). These results are shown in Appendix H.

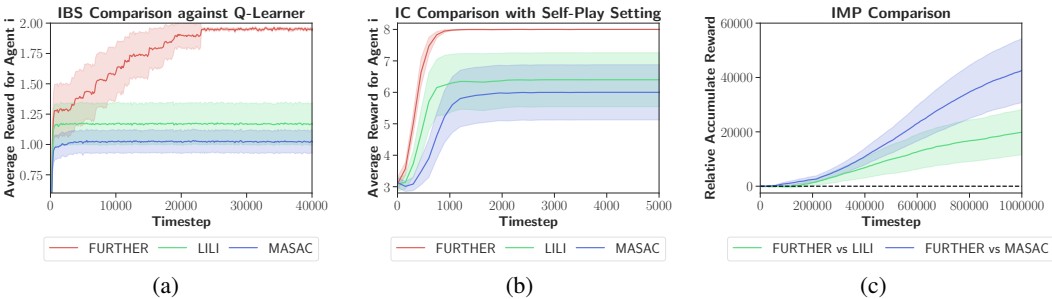

Figure 4: **(a)** Convergence in IBS. The FURTHER agent achieves convergence to its optimal pure strategy Nash equilibrium. **(b)** Convergence in IC with self-play. The FURTHER team shows better converged performance than baselines. **(c)** A competitive play in IMP between FURTHER and baseline methods. FURTHER receives higher rewards than LILI and MASAC over time.

**Question 1.** *How do methods perform when playing against a q-learning agent?*

We consider playing the iterated Bach or Stravinsky game (IBS; see Figure 3a). This general-sum game involves conflicting elements with two pure strategy Nash equilibria, where convergence to (B,B) and (S,S) equilibrium are more preferable from agent $i$'s and $j$'s perspective, respectively. Suppose agent $i$ plays against a naive learner $j$, such as $q$-learner [33], whose initial $q$-values are set to prefer action (S). In this experimental setting, it is ideal for agent $i$ to change $j$'s influence behavior to select (B) such that they converge to $i$'s optimal pure strategy Nash equilibrium of (B,B).

The average reward performance when an agent $i$, trained with either FURTHER or the baseline methods, interacts with the $q$-learner $j$ is shown in Figure 4a. There are two notable observations. First, the FURTHER agent $i$ consistently converges to its optimal equilibrium of (B,B), while the LILI agent often converges to the sub-optimal equilibrium of (S,S). The FURTHER agent $i$ learns to select (B) while $j$ selects (S), receives the worst rewards of zero, and waits until $j$'s $q$-value for (S) is updated to be lower than the $q$-value for (B). With the limiting view, $i$ learns that the waiting process is only temporary, and receiving the eventual rewards of 2 by converging to (B,B) is optimal. By contrast, LILI suffers from myopic evaluation and shows decreased performance upon convergence because the agent prefers simply converging to the sub-optimal equilibrium rather than waiting indefinitely. Figure 5a also shows that LILI achieves sub-optimal performance for any value of $\gamma$ and shows unstable learning as $\gamma \to 1$. Second, FURTHER and LILI outperform the other approach of MASAC, showing the benefit of considering the active influence on future policies of other agents.

**Question 2.** *Which equilibrium do methods converge to in a self-play setting?*

We experiment with a self-play setting in which both agents learn with the same algorithm in an iterated cooperative (IC) game with identical payoffs (see Figure 3b). This game has two pure strategy Nash equilibria of (U,U) and (D,D), in which the (D,D) equilibrium Pareto dominates the other. Figure 4b shows the average reward performance as the training iteration increases. First we find that LILI performs better than MASAC by considering the learning of agents. However, similar to the IBS results, we observe that FURTHER consistently converges to the best equilibrium of (D,D) while the baseline methods can converge to the sub-optimal equilibrium of (U,U) due to the myopic view.

**Question 3.** *How does FURTHER's limiting optimization perform directly against baselines?*

We consider FURTHER agent $i$ directly competing against either LILI or MASAC opponent $j$ in the iterated matching pennies (IMP) game (see Figure 3c). To show that FURTHER has a long-term perspective and thus can collect more rewards than the opposing method over time, we evaluate using a metric of relative accumulated reward summed up to the current timestep: $\sum_t r_t^i - r_t^j$. Figure 4c shows that the relative accumulated reward for FURTHER is positive for both settings, meaning that FURTHER receives higher rewards than LILI and MASAC over time. This result suggests that FURTHER is more effective than LILI by employing the limiting view via the average reward formulation. This result also conveys that it is beneficial to consider the underlying learning dynamics rather than ignoring them because FURTHER can more easily exploit the MASAC opponent and achieve higher accumulated rewards than when competing against the LILI opponent.

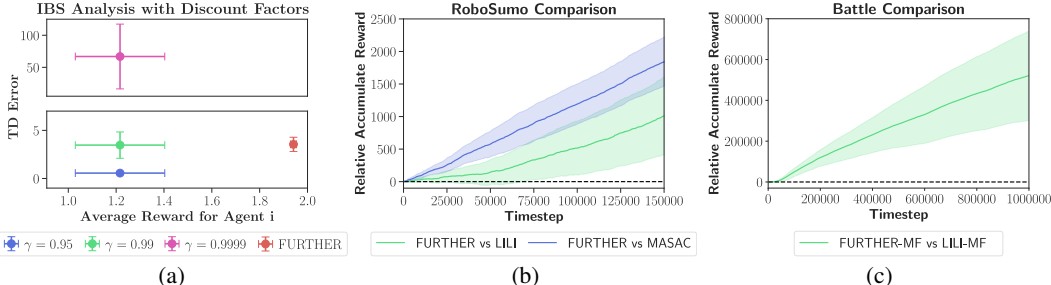

|  |  |  |
| :---: | :---: | :---: |
| (a) | (b) | (c) |

Figure 5: **(a)** Convergence performance and corresponding TD errors with varying $\gamma$ in LILI. As $\gamma \to 1$, LILI shows unstable learning (i.e., large TD error). **(b)** A competitive play in the RoboSumo domain, showing that FURTHER can learn a beneficial behavior in an environment with complex interactions **(c)** A mixed cooperative-competitive play in the battle domain. FURTHER-MF can solve a large-scale learning settings.

**Question 4.** *How does FURTHER scale to a more complex environment?*

To answer this question, we use the MuJoCo RoboSumo domain ([19]; see Figure 3d), where two ant robots compete with each other with the objective of pushing the opponent out of the ring. The reward function consists of a sparse reward of 5 for winning against the opponent and shaped rewards of moving towards the opponent and pushing the opponent further from the center of the ring. This environment has complex interactions because an agent must learn how to control its joints with continuous action space to move around the ring while learning to push the opponent. Similar to the setup in Question 3, FURTHER agent $i$ directly competes against either LILI or MASAC opponent $j$. We note that each agent has only partial observations about its opponent. As such, an agent infers its opponent's hidden policies and learning dynamics based on partial observations. We show the RoboSumo results in Figure 5b. Consistent with our results in the iterated matrix games, we observe that FURTHER gains more rewards than the baselines over time and wins against MASAC more often than against LILI. The averaged winning rate across the entire interaction shows that FURTHER wins against LILI and MASAC with 60.6% and 63.9%, respectively. Therefore, FURTHER provides a scalable framework that can learn policies in an environment with complex interactions and continuous action space.

**Question 5.** *How does FURTHER scale to a large number of agents?*

Finally, we show the scalability of our method regarding the number of agents using the battle domain ([32]; see Figure 3e). In this large-scale mixed cooperative-competitive setting, a red team of 25 agents and a blue team of 25 agents interact in a gridworld, where each agent collaborates with its teammates to eliminate the opponents. Specifically, we compare when red and blue agents learn with the mean-field version of FURTHER (i.e., FURTHER-MF) and LILI (i.e., LILI-MF), respectively, where they predict the mean actions of neighboring agents. We note that all 50 agents learn in a decentralized manner without sharing parameters with one another. It is evident that FURTHER-MF outperforms LILI-MF, which shows the effectiveness of having the limiting perspective. This result also conveys that FURTHER can easily incorporate other techniques in multiagent learning and show improved performance in large-scale settings.

## 5   Related Work

**Stationary MARL.**   The standard approach for addressing non-stationarity in MARL is to consider information about other agents and reason about joint action effects [34]. Example frameworks include centralized training with decentralized execution, which accounts for the actions of other agents through a centralized critic [12–14, 16, 35–37]. Other related approaches include opponent modeling frameworks that infer opponent policies and condition an agent's policy on this inferred information about others [38–41]. While this does alleviate non-stationarity, each agent learns its policy by assuming that other agents will follow the same policy into the future. This assumption is incorrect because other agents can have different behavior in the future due to their learning [6], resulting in instability with respect to their changing behavior. In contrast, FURTHER models the learning processes of other agents and considers how to actively influence limiting behavior.

**Learning-aware MARL.**    Our framework is closely related to prior work that considers the learning of other agents in the environment. The framework by [42], for instance, learns the best response adaptation to the other agent's anticipated updated policy. Notably, LOLA [6] and its more recent improvements [7, 43] study the impact of behavior on one or a few of another agent's policy updates. Our work is also related to frameworks that leverage the inferred policy dynamics of other agents to impact their future policies by maximizing the discounted return objective [8, 10]. Meta-learning frameworks are also related that directly account for the non-stationary policy dynamics in multiagent settings based on the inner-loop and outer-loop optimization [9, 11, 19, 44]. Lastly, the field of incentive MARL [45–48] is related, where agents additionally optimize incentive rewards and learn successful policies in solving sequential social dilemma domains [49, 50]. However, all of these approaches only account for a finite number of updates to the policies of other agents, so we observe that these methods can converge to a less desirable solution. FURTHER addresses this issue by optimizing for the average reward objective in the active Markov game setting.

**Game-theoretic MARL.**    Another effective approach to addressing the non-stationarity is learning equilibrium policies that correspond to game-theoretic solution concepts [4, 51–54]. These frameworks predict stationary joint action values by the end of learning and can guarantee convergence to Nash [3] or correlated [21] equilibrium values under certain assumptions. However, as noted in [55], this convergence is guaranteed only while ignoring the actual learning dynamics of other agents, and each agent assumes all agents will play the same joint equilibrium strategy. As such, equilibrium learners can fail to learn best-response policies when others choose to play different equilibrium strategies in the future as a result of their learning. By contrast, FURTHER considers convergence to a recurrent set of joint policies by inferring the true policy dynamics of other agents.

# 6    Conclusion

In this paper, we have introduced FURTHER to address non-stationarity by considering each agent's impact on the converged policies of other agents. The key idea is to consider the limiting policies of other agents through the average reward formulation for a newly proposed active Markov game framework, and we have developed a practical model-free and decentralized approach in this setting. We evaluated our method on various multiagent settings and showed that FURTHER consistently converges to more desirable long-term behavior than state-of-the-art baseline approaches.

### Acknowledgments

Research funded by IBM (as part of the MIT-IBM Watson AI Lab initiative).

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
