## A    Example of Initial Condition Sensitivity

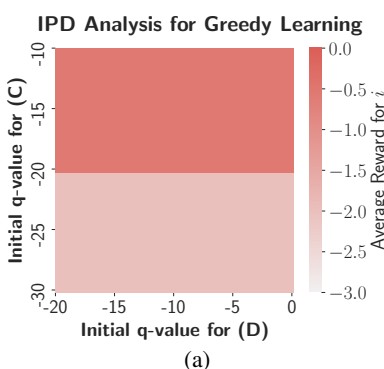
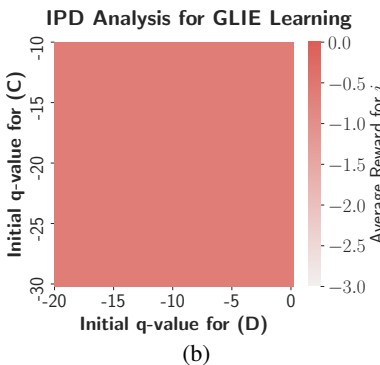

(a)                                              (b)

Figure 6: **(a)** A policy iteration analysis in IPD when agent $j$ has a greedy learning algorithm. Depending on $\boldsymbol{\theta}_0^{-i}$, $i$'s possible maximum average reward is affected. **(b)** A policy iteration analysis in IPD when agent $j$ has a GLIE learning algorithm. The possible maximum average reward for agent $i$ is independent to $j$'s initial policy $\boldsymbol{\theta}_0^{-i}$.

Consider playing the iterated prisoner's dilemma (IPD) game (see Table 1), where agent $i$ plays against a $q$-learning agent $j$. We perform a policy iteration analysis [17] with respect to $j$'s varying initial $q$-values for each action $\boldsymbol{\theta}_0^{-i}$. Figure 6a and Figure 6b show agent $i$'s average reward after convergence with respect to $\boldsymbol{\theta}_0^{i}$ when $j$ trains with a greedy and GLIE algorithm, respectively. Interestingly, the analysis with the greedy algorithm shows that $i$'s average reward depends on $\boldsymbol{\theta}_0^{-i}$ in IPD, where there is a set of $j$'s initial policies that $i$ can achieve the high average reward, but there is the other set of initial policies that can result in the undesirable average reward of –2. By contrast, Figure 6b shows that $i$'s average reward is independent of $\boldsymbol{\theta}_0^{-i}$ when $j$'s learning satisfies GLIE, empirically supporting our discussion in Section 2.3.

|   | $C$ | $D$ |
|---|---|---|
| $C$ | $(-1,-1)$ | $(-3,0)$ |
| $D$ | $(0,-3)$ | $(-2,-2)$ |

Table 1: Prisoner's dilemma game payoff matrix.

## B    Uniqueness of Jointly-Stable Periodic Distribution

**Proposition 1.** (Uniqueness of Jointly-Stable Periodic Distribution). *Given communicating state transition $\mathcal{T}$ and perturbed joint update function with noise $\mathcal{U}_\epsilon$, the jointly-stable periodic distribution is unique as $\epsilon \to 0$ over time.*

*Proof.* A perturbed Markov process has a unique stochastically stable distribution as noise $\epsilon \to 0$ over time if a perturbed Markov process is regular: the transition matrix corresponding to a stationary policy contains a single recurrent class of states (i.e., states that are visited infinitely often) and a possibly empty set of transient states (i.e., states that are visited only finitely often) [26] (Corollary 4.8, Section 5). As such, we prove that a Markov process of an active Markov game is regular by contradiction and thus show that the jointly-stable periodic distribution is unique as $\epsilon \to 0$. Suppose a perturbed Markov process of an active Markov game is irregular (i.e., there is more than one recurrent class), where the corresponding Markov matrix over the joint space of states and policies is defined as $p(s', \boldsymbol{\theta}'|s, \boldsymbol{\theta}) = \sum_{\boldsymbol{a}} \boldsymbol{\pi}(\boldsymbol{a}|s; \boldsymbol{\theta}) \mathcal{T}(s'|s, \boldsymbol{a}) \mathcal{U}_\epsilon(\boldsymbol{\theta}'|\boldsymbol{\theta}, \boldsymbol{\tau}) \; \forall s, s' \in \mathcal{S}, \boldsymbol{\theta}, \boldsymbol{\theta}' \in \Theta$. Because the perturbed joint update function has communicating strategies and thus contains a single recurrent class of policies, the state transition $\mathcal{T}$ must have multiple recurrent classes to result in an irregular active Markov game. However, $\mathcal{T}$ has a single recurrent class only due to the communicating assumption, which is the contradiction. Therefore, we conclude that a perturbed Markov process of an active game is regular, which has a unique stochastically stable distribution as $\epsilon \to 0$ by [26]. $\qquad\square$

## C   Derivation of Active Differential Bellman Equation

**Proposition 2.** (Active Differential Bellman Equation). *The differential value function $v^i_{\theta^i}$ represents the expected total difference between the accumulated rewards from $s$ and $\boldsymbol{\theta^{-i}}$ and the average reward $\rho^i_{\theta^i}$ [18]. The differential value function inherently includes the recursive relationship with respect to $v^i_{\theta^i}$ at the next state $s'$ and the updated policies of other agents $\boldsymbol{\theta^{-i\prime}}$:*

$$
v^i_{\theta^i}(s, \boldsymbol{\theta^{-i}}) = \lim_{T\to\infty} \mathbb{E}\Big[ \sum_{t=0}^{T} \big(\mathcal{R}^i(s_t, \boldsymbol{a_t}) - \rho^i_{\theta^i}\big) \Big|\, {\substack{s_0=s,\ \boldsymbol{\theta_0^{-i}}=\boldsymbol{\theta^{-i}},\\ a_{0:T}^i\sim\pi(\cdot|s_{0:T};\theta^i),\boldsymbol{a_{0:T}^{-i}}\sim\boldsymbol{\pi}(\cdot|s_{0:T};\boldsymbol{\theta_{0:T}^{-i}}),\\ s_{t+1}\sim\mathcal{T}(\cdot|s_t,\boldsymbol{a_t}),\boldsymbol{\theta_{t+1}^{-i}}\sim\boldsymbol{\mathcal{U}^{-i}}(\cdot|\boldsymbol{\theta_t^{-i}},\boldsymbol{\tau_t^{-i}})}} \Big]
$$

$$
= \sum_{a^i} \pi(a^i|s;\theta^i) \sum_{\boldsymbol{a^{-i}}} \boldsymbol{\pi}(\boldsymbol{a^{-i}}|s;\boldsymbol{\theta^{-i}}) \sum_{s'} \mathcal{T}(s'|s,\boldsymbol{a}) \sum_{\boldsymbol{\theta^{-i\prime}}} \boldsymbol{\mathcal{U}^{-i}}(\boldsymbol{\theta^{-i\prime}}|\boldsymbol{\theta^{-i}}, \boldsymbol{\tau^{-i}})
$$

$$
\Big[\mathcal{R}^i(s, \boldsymbol{a}) - \rho^i_{\theta^i} + v^i_{\theta^i}(s', \boldsymbol{\theta^{-i\prime}})\Big].
$$

*Proof.* We seek to derive the recursive relationship between $v^i_{\theta^i}(s, \boldsymbol{\theta^{-i}})$ and $v^i_{\theta^i}(s', \boldsymbol{\theta^{-i\prime}})$. We leverage the general derivation outlined in [18] (page 59) and extend it to our active Markov game formulation:

$$
v^i_{\theta^i}(s, \boldsymbol{\theta^{-i}}) = \lim_{T\to\infty} \mathbb{E}\Big[ \sum_{t=0}^{T} \big(\mathcal{R}^i(s_t, \boldsymbol{a_t}) - \rho^i_{\theta^i}\big) \Big|\, {\substack{s_0=s,\ \boldsymbol{\theta_0^{-i}}=\boldsymbol{\theta^{-i}},\\ a_{0:T}^i\sim\pi(\cdot|s_{0:T};\theta^i),\boldsymbol{a_{0:T}^{-i}}\sim\boldsymbol{\pi}(\cdot|s_{0:T};\boldsymbol{\theta_{0:T}^{-i}}),\\ s_{t+1}\sim\mathcal{T}(\cdot|s_t,\boldsymbol{a_t}),\boldsymbol{\theta_{t+1}^{-i}}\sim\boldsymbol{\mathcal{U}^{-i}}(\cdot|\boldsymbol{\theta_t^{-i}},\boldsymbol{\tau_t^{-i}})}} \Big]
$$

$$
= \lim_{T\to\infty} \mathbb{E}\Big[ \mathcal{R}^i(s_0, \boldsymbol{a_0}) - \rho^i_{\theta^i} + \sum_{t=1}^{T} \big(\mathcal{R}^i(s_t, \boldsymbol{a_t^{-i}}) - \rho^i_{\theta^i}\big) \Big|\, {\substack{s_0=s,\ \boldsymbol{\theta_0^{-i}}=\boldsymbol{\theta^{-i}},\\ a_{0:T}^i\sim\pi(\cdot|s_{0:T};\theta^i),\boldsymbol{a_{0:T}^{-i}}\sim\boldsymbol{\pi}(\cdot|s_{0:T};\boldsymbol{\theta_{0:T}^{-i}}),\\ s_{t+1}\sim\mathcal{T}(\cdot|s_t,\boldsymbol{a_t}),\boldsymbol{\theta_{t+1}^{-i}}\sim\boldsymbol{\mathcal{U}^{-i}}(\cdot|\boldsymbol{\theta_t^{-i}},\boldsymbol{\tau_t^{-i}})}} \Big]
$$

$$
= \sum_{a^i} \pi(a^i|s;\theta^i) \sum_{\boldsymbol{a^{-i}}} \boldsymbol{\pi}(\boldsymbol{a^{-i}}|s;\boldsymbol{\theta^{-i}}) \sum_{s'} \mathcal{T}(s'|s,\boldsymbol{a}) \sum_{\boldsymbol{\theta^{-i\prime}}} \boldsymbol{\mathcal{U}^{-i}}(\boldsymbol{\theta^{-i\prime}}|\boldsymbol{\theta^{-i}}, \boldsymbol{\tau^{-i}})
$$

$$
\Big[\mathcal{R}^i(s, \boldsymbol{a}) - \rho^i_{\theta^i} + \lim_{T\to\infty} \mathbb{E}\Big[\sum_{t=0}^{T}\big(\mathcal{R}^i(s_{t+1}, \boldsymbol{a_{t+1}}) - \rho^i_{\theta^i}\big) \Big|\, {\substack{s_1=s',\ \boldsymbol{\theta_1^{-i}}=\boldsymbol{\theta^{-i\prime}},\\ a_{1:T}^i\sim\pi(\cdot|s_{1:T};\theta^i),\boldsymbol{a_{1:T}^{-i}}\sim\boldsymbol{\pi}(\cdot|s_{1:T};\boldsymbol{\theta_{1:T}^{-i}}),\\ s_{t+1}\sim\mathcal{T}(\cdot|s_t,\boldsymbol{a_t}),\boldsymbol{\theta_{t+1}^{-i}}\sim\boldsymbol{\mathcal{U}^{-i}}(\cdot|\boldsymbol{\theta_t^{-i}},\boldsymbol{\tau_t^{-i}})}} \Big]\Big]
$$

$$
= \sum_{a^i} \pi(a^i|s;\theta^i) \sum_{\boldsymbol{a^{-i}}} \boldsymbol{\pi}(\boldsymbol{a^{-i}}|s;\boldsymbol{\theta^{-i}}) \sum_{s'} \mathcal{T}(s'|s,\boldsymbol{a}) \sum_{\boldsymbol{\theta^{-i\prime}}} \boldsymbol{\mathcal{U}^{-i}}(\boldsymbol{\theta^{-i\prime}}|\boldsymbol{\theta^{-i}}, \boldsymbol{\tau^{-i}})
$$

$$
\Big[\mathcal{R}^i(s, \boldsymbol{a}) - \rho^i_{\theta^i} + v^i_{\theta^i}(s', \boldsymbol{\theta^{-i\prime}})\Big]. \tag{15}
$$

$\square$

## D   Derivation of Active Average Reward Policy Gradient

**Proposition 3.** (Active Average Reward Policy Gradient Theorem). *The gradient of active average reward objective in Equation (7) with respect to agent $i$'s policy parameters $\theta^i$ is:*

$$
\nabla_{\theta^i} J^i_\pi(\theta^i) = \frac{1}{k}\sum_{\ell=1}^{k} \sum_{s_\ell, \boldsymbol{\theta_\ell^{-i}}} \mu_{k,\theta^i}(s_\ell, \boldsymbol{\theta_\ell}|\ell) \sum_{a_\ell^i} \nabla_{\theta^i}\pi(a_\ell^i|s_\ell;\theta^i) \sum_{\boldsymbol{a_\ell^{-i}}} \boldsymbol{\pi}(\boldsymbol{a_\ell^{-i}}|s_\ell;\boldsymbol{\theta_\ell^{-i}}) q^i_{\theta^i}(s_\ell, \boldsymbol{\theta_\ell^{-i}}, \boldsymbol{a_\ell}),
$$

*with* $q^i_{\theta^i}(s_\ell, \boldsymbol{\theta_\ell^{-i}}, \boldsymbol{a_\ell}) = \sum_{s_{\ell+1}} \mathcal{T}(s_{\ell+1}|s_\ell, \boldsymbol{a_\ell}) \sum_{\boldsymbol{\theta_{\ell+1}^{-i}}} \boldsymbol{\mathcal{U}^{-i}}(\boldsymbol{\theta_{\ell+1}^{-i}}|\boldsymbol{\theta_\ell^{-i}}, \boldsymbol{\tau_\ell^{-i}})\Big[\mathcal{R}^i(s_\ell, \boldsymbol{a_\ell}) - \rho^i_{\theta^i} + v^i_{\theta^i}(s_{\ell+1}, \boldsymbol{\theta_{\ell+1}^{-i}})\Big].$

*Proof.* We seek to derive an expression for optimizing the average reward objective in Equation (7) with respect to agent $i$'s policy parameters $\theta^i$. Our derivation leverages the general policy gradient theorem proof for the continuing case in [18] (page 334). We begin by expressing the gradient of the differential value function $v^i_{\theta^i}(s, \boldsymbol{\theta^{-i}})$ for $s \in \mathcal{S}$ and $\boldsymbol{\theta^{-i}} \in \boldsymbol{\Theta^{-i}}$:

$$
\nabla_{\theta^i} v^i_{\theta^i}(s, \boldsymbol{\theta^{-i}}) = \nabla_{\theta^i}\Big[ \sum_{a^i} \pi(a^i|s;\theta^i) \sum_{\boldsymbol{a^{-i}}} \boldsymbol{\pi}(\boldsymbol{a^{-i}}|s;\boldsymbol{\theta^{-i}}) q^i_{\theta^i}(s, \boldsymbol{\theta^{-i}}, \boldsymbol{a})\Big]
$$

$$
= \sum_{a^i} \nabla_{\theta^i}\pi(a^i|s;\theta^i) \sum_{\boldsymbol{a^{-i}}} \boldsymbol{\pi}(\boldsymbol{a^{-i}}|s;\boldsymbol{\theta^{-i}}) q^i_{\theta^i}(s, \boldsymbol{\theta^{-i}}, \boldsymbol{a}) +
$$

$$
\sum_{a^i} \pi(a^i|s;\theta^i) \sum_{\boldsymbol{a^{-i}}} \boldsymbol{\pi}(\boldsymbol{a^{-i}}|s;\boldsymbol{\theta^{-i}}) \underbrace{\nabla_{\theta^i} q^i_{\theta^i}(s, \boldsymbol{\theta^{-i}}, \boldsymbol{a})}_{\text{Term A}}. \tag{16}
$$

We continue to derive the Term A in Equation (16):

$$\nabla_{\theta^i} q^i_{\theta^i}(s, \boldsymbol{\theta}^{-i}, \boldsymbol{a}) = \nabla_{\theta^i}\left[\sum_{s'}\mathcal{T}(s'|s,\boldsymbol{a})\sum_{\boldsymbol{\theta}^{-i\prime}}\mathcal{U}^{-i}(\boldsymbol{\theta}^{-i\prime}|\boldsymbol{\theta}^{-i},\boldsymbol{\tau}^{-i})\Big[\mathcal{R}^i(s,\boldsymbol{a})-\rho^i_{\theta^i}+v^i_{\theta^i}(s',\boldsymbol{\theta}^{-i\prime})\Big]\right]$$

$$= -\nabla_{\theta^i}\rho^i_{\theta^i} + \sum_{s'}\mathcal{T}(s'|s,\boldsymbol{a})\sum_{\boldsymbol{\theta}^{-i\prime}}\mathcal{U}^{-i}(\boldsymbol{\theta}^{-i\prime}|\boldsymbol{\theta}^{-i},\boldsymbol{\tau}^{-i})\nabla_{\theta^i}v^i_{\theta^i}(s',\boldsymbol{\theta}^{-i\prime}). \qquad (17)$$

We summarize Equation (16) and Equation (17) together and re-arrange terms to obtain:

$$\nabla_{\theta^i}\rho^i_{\theta^i} = \sum_{a^i}\nabla_{\theta^i}\pi(a^i|s;\theta^i)\sum_{\boldsymbol{a}^{-i}}\boldsymbol{\pi}(\boldsymbol{a}^{-i}|s;\boldsymbol{\theta}^{-i})q^i_{\theta^i}(s,\boldsymbol{\theta}^{-i},\boldsymbol{a})+$$

$$\sum_{a^i}\pi(\cdot|s;\theta^i)\sum_{\boldsymbol{a}^{-i}}\boldsymbol{\pi}(\boldsymbol{a}^{-i}|s;\boldsymbol{\theta}^{-i})\sum_{s'}\mathcal{T}(s'|s,\boldsymbol{a})\sum_{\boldsymbol{\theta}^{-i\prime}}\mathcal{U}^{-i}(\boldsymbol{\theta}^{-i\prime}|\boldsymbol{\theta}^{-i},\boldsymbol{\tau}^{-i})\nabla_{\theta^i}v^i_{\theta^i}(s',\boldsymbol{\theta}^{-i\prime})-$$

$$\nabla_{\theta^i}v^i_{\theta^i}(s,\boldsymbol{\theta}^{-i}). \qquad (18)$$

We define the jointly-stable periodic distribution with respect to the agent $i$'s fixed stationary policy:

$$\frac{1}{k}\sum_{\ell=1}^{k}\mu_{k,\theta^i}(s_{\ell+1},\boldsymbol{\theta}_{\ell+1}|\ell+1) = \frac{1}{k}\sum_{\ell=1}^{k}\sum_{s_\ell,\boldsymbol{\theta}_\ell^{-i}}\mu_{k,\theta^i}(s_\ell,\boldsymbol{\theta}_\ell|\ell)\sum_{\boldsymbol{a}_\ell}\boldsymbol{\pi}(\boldsymbol{a}_\ell|s_\ell;\boldsymbol{\theta}_\ell)$$

$$\mathcal{T}(s_{\ell+1}|s_\ell,\boldsymbol{a}_\ell)\,\mathcal{U}^{-i}(\boldsymbol{\theta}_{\ell+1}^{-i}|\boldsymbol{\theta}_\ell^{-i},\boldsymbol{\tau}_\ell^{-i}) \quad \forall s_{\ell+1}\in\mathcal{S},\boldsymbol{\theta}_{\ell+1}\in\boldsymbol{\Theta}, \qquad (19)$$

where $\boldsymbol{\theta}_\ell = \{\theta^i, \boldsymbol{\theta}_\ell^{-i}\}$. We now apply Equation (19) to Equation (18) and derive the final expression for policy gradient by writing $\nabla_{\theta^i}\rho^i_{\theta^i}$ as $\nabla_{\theta^i}J^i_\pi(\theta^i)$:

$$\frac{1}{k}\sum_{\ell=1}^{k}\sum_{s_\ell,\boldsymbol{\theta}_\ell^{-i}}\mu_{k,\theta^i}(s_\ell,\boldsymbol{\theta}_\ell|\ell)\nabla_{\theta^i}J^i_\pi(\theta^i) = \frac{1}{k}\sum_{\ell=1}^{k}\sum_{s_\ell,\boldsymbol{\theta}_\ell^{-i}}\mu_{k,\theta^i}(s_\ell,\boldsymbol{\theta}_\ell|\ell)\Bigg[$$

$$\sum_{a_\ell^i}\nabla_{\theta^i}\pi(a_\ell^i|s_\ell;\theta^i)\sum_{\boldsymbol{a}_\ell^{-i}}\boldsymbol{\pi}(\boldsymbol{a}_\ell^{-i}|s_\ell;\boldsymbol{\theta}_\ell^{-i})q^i_{\theta^i}(s_\ell,\boldsymbol{\theta}_\ell^{-i},\boldsymbol{a}_\ell)+$$

$$\sum_{a_\ell^i}\pi(a_\ell^i|s_\ell;\theta^i)\sum_{\boldsymbol{a}_\ell^{-i}}\boldsymbol{\pi}(\boldsymbol{a}_\ell^{-i}|s_\ell;\boldsymbol{\theta}_\ell^{-i})\sum_{s_{\ell+1}}\mathcal{T}(s_{\ell+1}|s_\ell,\boldsymbol{a}_\ell)\sum_{\boldsymbol{\theta}_{\ell+1}^{-i}}\mathcal{U}^{-i}(\boldsymbol{\theta}_{\ell+1}^{-i}|\boldsymbol{\theta}_\ell^{-i},\boldsymbol{\tau}_\ell^{-i})\nabla_{\theta^i}v^i_{\theta^i}(s_{\ell+1},\boldsymbol{\theta}_{\ell+1}^{-i})-$$

$$\nabla_{\theta^i}v^i_{\theta^i}(s_\ell,\boldsymbol{\theta}_\ell^{-i})\Bigg]. \qquad (20)$$

Note that the left-hand side $\nabla_{\theta^i}J^i_\pi(\theta^i)$ does not depend on $s_\ell$ and $\boldsymbol{\theta}_\ell^{-i}$, so Equation (20) becomes:

$$\nabla_{\theta^i}J^i_\pi(\theta^i)$$

$$= \frac{1}{k}\sum_{\ell=1}^{k}\sum_{s_\ell,\boldsymbol{\theta}_\ell^{-i}}\mu_{k,\theta^i}(s_\ell,\boldsymbol{\theta}_\ell|\ell)\sum_{a_\ell^i}\nabla_{\theta^i}\pi(a_\ell^i|s_\ell;\theta^i)\sum_{\boldsymbol{a}_\ell^{-i}}\boldsymbol{\pi}(\boldsymbol{a}_\ell^{-i}|s_\ell;\boldsymbol{\theta}_\ell^{-i})q^i_{\theta^i}(s_\ell,\boldsymbol{\theta}_\ell^{-i},\boldsymbol{a}_\ell)+$$

$$\frac{1}{k}\sum_{\ell=1}^{k}\sum_{s_\ell,\boldsymbol{\theta}_\ell^{-i}}\mu_{k,\theta^i}(s_\ell,\boldsymbol{\theta}_\ell|\ell)\sum_{a_\ell^i}\pi(a_\ell^i|s_\ell;\theta^i)\sum_{\boldsymbol{a}_\ell^{-i}}\boldsymbol{\pi}(\boldsymbol{a}_\ell^{-i}|s_\ell;\boldsymbol{\theta}_\ell^{-i})\sum_{s_{\ell+1}}\mathcal{T}(s_{\ell+1}|s_\ell,\boldsymbol{a}_\ell)$$

$$\sum_{\boldsymbol{\theta}_{\ell+1}^{-i}}\mathcal{U}^{-i}(\boldsymbol{\theta}_{\ell+1}^{-i}|\boldsymbol{\theta}_\ell^{-i},\boldsymbol{\tau}_\ell^{-i})\nabla_{\theta^i}v^i_{\theta^i}(s_{\ell+1},\boldsymbol{\theta}_{\ell+1}^{-i})-$$

$$\frac{1}{k}\sum_{\ell=1}^{k}\sum_{s_\ell,\boldsymbol{\theta}_\ell^{-i}}\mu_{k,\theta^i}(s_\ell,\boldsymbol{\theta}_\ell|\ell)\nabla_{\theta^i}v^i_{\theta^i}(s_\ell,\boldsymbol{\theta}_\ell^{-i})$$

$$= \frac{1}{k}\sum_{\ell=1}^{k}\sum_{s_\ell,\boldsymbol{\theta}_\ell^{-i}}\mu_{k,\theta^i}(s_\ell,\boldsymbol{\theta}_\ell|\ell)\sum_{a_\ell^i}\nabla_{\theta^i}\pi(a_\ell^i|s_\ell;\theta^i)\sum_{\boldsymbol{a}_\ell^{-i}}\boldsymbol{\pi}(\boldsymbol{a}_\ell^{-i}|s_\ell;\boldsymbol{\theta}_\ell^{-i})q^i_{\theta^i}(s_\ell,\boldsymbol{\theta}_\ell^{-i},\boldsymbol{a}_\ell)+$$

$$\frac{1}{k}\sum_{\ell=1}^{k}\sum_{\substack{s_{\ell+1},\\\boldsymbol{\theta}_{\ell+1}^{-i}}}\mu_{k,\theta^i}(s_{\ell+1},\boldsymbol{\theta}_{\ell+1}|\ell+1)\nabla_{\theta^i}v^i_{\theta^i}(s_{\ell+1},\boldsymbol{\theta}_{\ell+1}^{-i})-\frac{1}{k}\sum_{\ell=1}^{k}\sum_{s_\ell,\boldsymbol{\theta}_\ell^{-i}}\mu_{k,\theta^i}(s_\ell,\boldsymbol{\theta}_\ell|\ell)\nabla_{\theta^i}v^i_{\theta^i}(s_\ell,\boldsymbol{\theta}_\ell^{-i})$$

$$= \frac{1}{k}\sum_{\ell=1}^{k}\sum_{s_\ell,\boldsymbol{\theta}_\ell^{-i}}\mu_{k,\theta^i}(s_\ell,\boldsymbol{\theta}_\ell|\ell)\sum_{a_\ell^i}\nabla_{\theta^i}\pi(a_\ell^i|s_\ell;\theta^i)\sum_{\boldsymbol{a}_\ell^{-i}}\boldsymbol{\pi}(\boldsymbol{a}_\ell^{-i}|s_\ell;\boldsymbol{\theta}_\ell^{-i})q^i_{\theta^i}(s_\ell,\boldsymbol{\theta}_\ell^{-i},\boldsymbol{a}_\ell). \qquad (21)$$

$$\square$$

# E  Additional Implementation Details

## E.1  Network Structure

Our neural networks for the inference learning and reinforcement learning module consist of fully-connected layers for vector observations (e.g., iterated matrix games, MuJoCo RoboSumo [19]) and additional convolution layers for image observations (e.g., MAgent Battle [32]). The encoder outputs the mean and standard deviation for the Gaussian distribution of $p(\hat{z}_{t+1}^{-i}|\hat{z}_t^{-i}, \tau_t^i; \phi_{\text{enc}}^i)$, where we sample $\hat{z}_t^{-i}$ by applying the reparameterization trick [28]. From the sampled $\hat{z}_t^{-i}$, the decoder $p(a_t^{-i}|s_t, \hat{z}_t^{-i}; \phi_{\text{dec}}^i)$ outputs a probability for the categorical distribution (discrete action space) or a mean and variance for the Gaussian distribution (continuous action space). Similarly, the policy $\pi(a_t^i|s_t, \hat{z}_t^{-i}; \theta^i)$ outputs a probability for the categorical distribution (discrete action space) or a mean and variance for the Gaussian distribution (continuous action space). Lastly, the critic outputs $q$-values for all actions for discrete action space (i.e., $q_{\theta^i}^i(a_t^i|s_t, \hat{z}_t^{-i}, a_t^{-i}; \psi_\beta^i)$) by following [31] or outputs a $q$-value given the joint action for continuous action space (i.e., $q_{\theta^i}^i(s_t, \hat{z}_t^{-i}, a_t; \psi_\beta^i)$).

## E.2  Optimization

We detail additional notes about our implementation:

• For simplicity, we consider the period $k=1$ and develop corresponding soft reinforcement learning optimizations in Section 3.2. The current FURTHER implementation can be extended to settings with $k>1$ by sampling $k$ states and policies that are consecutive within each batch.

• For continuous action space, we modify SAC for continuous action space [27] and replace the soft value function $v_{\theta^i}^i$ in Equation (13) with:

$$v_{\theta^i}^i(s, \hat{z}^{-i}; \psi^i) = \mathbb{E}_{a^i \sim \pi(\cdot|s, \hat{z}^{-i}; \theta^i), a^{-i} \sim \pi(\cdot|s; \hat{z}^{-i})}\Big[\min_{\beta=1,2} q_{\theta^i}^i(s, \hat{z}^{-i}, a; \psi_\beta^i)\Big] + \alpha \mathcal{H}(\pi(\cdot|s, \hat{z}^{-i}; \theta^i)). \quad (22)$$

We also replace the policy optimization in Equation (14) with the following:

$$J_\pi^i(\theta^i) = \mathbb{E}_{(s, \hat{z}^{-i}, a^{-i}) \sim \mathcal{D}^i, \epsilon \sim \mathcal{N}(0, I)}\Big[$$
$$\min_{\beta=1,2} q_{\theta^i}^i(s, \hat{z}^{-i}, f_{\theta^i}(\epsilon; s, \hat{z}^{-i}), a^{-i}; \psi_\beta^i) - \alpha \log \pi(f_{\theta^i}(\epsilon; s, \hat{z}^{-i})|s, \hat{z}^{-i}; \theta^i)\Big], \quad (23)$$

where $a^i = f_{\theta^i}(\epsilon; s, \hat{z}^{-i})$ denotes the output of the reparameterized $i$'s policy [27].

• In practice, we apply a weighting of $0.01$ on the KL divergence term in Equation (11) for balanced training of the inference learning module.

• Because it is impractical to consider the entire interactions from the beginning of the game in computing Equation (11), we limit $\tau_{0:t-1}^i$ to be recent interactions specified by a batch size.

### E.3 Pseudocode

---
**Algorithm 1** FURTHER and FURTHER Mean-Field

---
**Require:** Learning rates $\alpha_q, \alpha_\rho, \alpha_\pi, \alpha_\phi$, soft $q$-target update rate $\tau_q$
1: *# Agent initialization*
2: **for** Each agent $i$ **do**
3:     Initialize RL module $\theta^i, \psi_1^i, \psi_2^i, \bar{\psi}_1^i, \bar{\psi}_2^i, \rho_{\theta^i}^i, \mathcal{D}^i$
4:     Initialize inference module $\phi_{\text{enc}}^i, \phi_{\text{dec}}^i$
5:     Initialize other agents' latent strategies $\hat{\boldsymbol{z}}_{\mathbf{0}}^{\boldsymbol{-i}}$
6: **end for**
7: **for** Each timestep $t$ **do**
8:     *# Decentralized execution*
9:     **for** Each agent $i$ **do**
10:        Select action $a_t^i \sim \pi(\cdot|s_t, \hat{\boldsymbol{z}}_{\boldsymbol{t}}^{\boldsymbol{-i}}; \theta^i)$
11:     **end for**
12:     Execute joint action $\boldsymbol{a_t}$ and receive next state $s_{t+1}$ and joint rewards $\boldsymbol{r_t}$
13:     *# Mean action computation and perform inference*
14:     **for** Each agent $i$ **do**
15:        **if** Apply mean-field **then**
16:           Compute mean action of its neighborhood $\bar{a}_t^{-i}$ and set $\boldsymbol{a_t} = \{a_t^i, \bar{a}_t^{-i}\}$
17:        **end if**
18:        Infer next updated policies of other agents $\hat{\boldsymbol{z}}_{\boldsymbol{t+1}}^{\boldsymbol{-i}} \sim p(\cdot|\hat{\boldsymbol{z}}_{\boldsymbol{t}}^{\boldsymbol{-i}}, \tau_t^i; \phi_{\text{enc}}^i)$
19:        Add a transition to its replay memory $\mathcal{D}^i \leftarrow \mathcal{D}^i \cup \{s_t, \hat{\boldsymbol{z}}_{\boldsymbol{t}}^{\boldsymbol{-i}}, \boldsymbol{a_t}, r_t^i, s_{t+1}, \hat{\boldsymbol{z}}_{\boldsymbol{t+1}}^{\boldsymbol{-i}}\}$
20:     **end for**
21:     *# Decentralized training*
22:     **for** Each agent $i$ **do**
23:        $\{\psi_\beta^i, \rho_{\theta^i}^i\} \leftarrow \{\psi_\beta^i, \rho_{\theta^i}^i\} - \{\alpha_q, \alpha_\rho\} J_q^i(\psi_\beta^i, \rho_{\theta^i})$ for $\beta = 1, 2$
24:        $\theta^i \leftarrow \theta^i + \alpha_\pi J_\pi^i(\theta^i)$
25:        $\{\phi_{\text{enc}}^i, \phi_{\text{dec}}^i\} \leftarrow \{\phi_{\text{enc}}^i, \phi_{\text{dec}}^i\} - \alpha_\phi J_{\text{elbo}}^i(\phi_{\text{enc}}^i, \phi_{\text{dec}}^i)$
26:        $\bar{\psi}_\beta^i \leftarrow \tau_q \psi_\beta^i + (1 - \tau_q)\bar{\psi}_\beta^i$ for $\beta = 1, 2$
27:     **end for**
28: **end for**

---

## F   ELBO Derivation

We derive our ELBO optimization in Equation (11) for the inference module. In particular, we follow the ELBO derivation in [56] (Appendix A) and modify it for our multiagent setting:

$$
\begin{aligned}
\mathbb{E}_{p(\tau_{0:t}^i)}\Big[\log p(\tau_{1:t}^i; \phi_{\text{dec}}^i)\Big] &= \mathbb{E}_{p(\tau_{0:t}^i)}\Big[\log \int p(\tau_{1:t}^i, \hat{\boldsymbol{z}}_{\boldsymbol{1:t}}^{\boldsymbol{-i}}; \phi_{\text{dec}}^i) d\hat{\boldsymbol{z}}_{\boldsymbol{1:t}}^{\boldsymbol{-i}}\Big] \\
&= \mathbb{E}_{p(\tau_{0:t}^i)}\Big[\log \int p(\tau_{1:t}^i, \hat{\boldsymbol{z}}_{\boldsymbol{1:t}}^{\boldsymbol{-i}}; \phi_{\text{dec}}^i) \frac{p(\hat{\boldsymbol{z}}_{\boldsymbol{1:t}}^{\boldsymbol{-i}}|\tau_{0:t-1}^i; \phi_{\text{enc}}^i)}{p(\hat{\boldsymbol{z}}_{\boldsymbol{1:t}}^{\boldsymbol{-i}}|\tau_{0:t-1}^i; \phi_{\text{enc}}^i)} d\hat{\boldsymbol{z}}_{\boldsymbol{1:t}}^{\boldsymbol{-i}}\Big] \\
&= \mathbb{E}_{p(\tau_{0:t}^i)}\Big[\log \mathbb{E}_{p(\hat{\boldsymbol{z}}_{\boldsymbol{1:t}}^{\boldsymbol{-i}}|\tau_{0:t-1}^i; \phi_{\text{enc}}^i)}\big[\frac{p(\tau_{1:t}^i, \hat{\boldsymbol{z}}_{\boldsymbol{1:t}}^{\boldsymbol{-i}}; \phi_{\text{dec}}^i)}{p(\hat{\boldsymbol{z}}_{\boldsymbol{1:t}}^{\boldsymbol{-i}}|\tau_{0:t-1}^i; \phi_{\text{enc}}^i)}\big]\Big] \\
&\geq \mathbb{E}_{p(\tau_{0:t}^i), p(\hat{\boldsymbol{z}}_{\boldsymbol{1:t}}^{\boldsymbol{-i}}|\tau_{0:t-1}^i; \phi_{\text{enc}}^i)}\Big[\log \frac{p(\tau_{1:t}^i, \hat{\boldsymbol{z}}_{\boldsymbol{1:t}}^{\boldsymbol{-i}}; \phi_{\text{dec}}^i)}{p(\hat{\boldsymbol{z}}_{\boldsymbol{1:t}}^{\boldsymbol{-i}}|\tau_{0:t-1}^i; \phi_{\text{enc}}^i)}\Big] \\
&= \mathbb{E}_{p(\tau_{0:t}^i), p(\hat{\boldsymbol{z}}_{\boldsymbol{1:t}}^{\boldsymbol{-i}}|\tau_{0:t-1}^i; \phi_{\text{enc}}^i)}\Big[\log p(\tau_{1:t}^i, \hat{\boldsymbol{z}}_{\boldsymbol{1:t}}^{\boldsymbol{-i}}; \phi_{\text{dec}}^i) - \log p(\hat{\boldsymbol{z}}_{\boldsymbol{1:t}}^{\boldsymbol{-i}}|\tau_{0:t-1}^i; \phi_{\text{enc}}^i)\Big] \\
&= \mathbb{E}_{p(\tau_{0:t}^i), p(\hat{\boldsymbol{z}}_{\boldsymbol{1:t}}^{\boldsymbol{-i}}|\tau_{0:t-1}^i; \phi_{\text{enc}}^i)}\Big[\sum_{t'=1}^{t} \log p(\boldsymbol{a}_{\boldsymbol{t'}}^{\boldsymbol{-i}}|s_{t'}, \hat{\boldsymbol{z}}_{\boldsymbol{t'}}^{\boldsymbol{-i}}; \phi_{\text{dec}}^i) + \sum_{t'=0}^{t-1} \log p(\hat{\boldsymbol{z}}_{\boldsymbol{t'}}^{\boldsymbol{-i}}) - \\
&\qquad\qquad \sum_{t'=1}^{t} \log p(\hat{\boldsymbol{z}}_{\boldsymbol{t'}}^{\boldsymbol{-i}}|\tau_{t'-1}^i; \phi_{\text{enc}}^i)\Big].
\end{aligned} \tag{24}
$$

Finally, we summarize terms to obtain:

$$\mathbb{E}_{p(\tau_{0:t}^i), p(\hat{\boldsymbol{z}}_{1:t}^{-i}|\tau_{0:t-1}^i; \phi_{\text{enc}}^i)} \Big[ \sum_{t'=1}^{t} \underbrace{\log p(\boldsymbol{a}_{t'}^{-i}|s_{t'}, \hat{\boldsymbol{z}}_{t'}^{-i}; \phi_{\text{dec}}^i)}_{\text{Reconstruction loss}} - \underbrace{D_{\text{KL}}\big(p(\hat{\boldsymbol{z}}_{t'}^{-i}|\tau_{t'-1}^i; \phi_{\text{enc}}^i) || p(\hat{\boldsymbol{z}}_{t'-1}^{-i})\big)}_{\text{KL divergence}} \Big].$$

## G   Experimental and Hyperparameter Details

### G.1   Domain Details

**Iterated matrix games.**   As in [6], we model the state space in all iterated matrix games as $s_0 = \varnothing$ and $s_t = \boldsymbol{a_{t-1}}$ for $t \geq 1$. For these simple domains, we empirically observe that training the policy and critics based on the most recent transition improves training performance. Lastly, in Question 1, we consider agent $i$ playing against a $q$-learning agent $j$ with a learning rate $\alpha_q$ of 0.5, a discount factor $\gamma$ of 0.9, and a fixed $\epsilon$-exploration of 0.05.

**MuJoco RoboSumo.**   Each ant robot observes a vector with size 128, which consists of the position of its own and the opponent's body, its own joint angles and velocities, and forces exerted on each part of its own body and the opponent's torso [19]. We note that each agent has partial observations about its opponent and cannot observe the opponent's velocities and limb positions. Regarding the action space, each agent has a continuous action space with a dimension of 8. Lastly, we use the reward function that consists of a sparse reward of 5 for winning against the opponent and the following shaped rewards:

• Reward for moving towards the opponent proportional to $-d_{\text{opp}}$, where $d_{\text{opp}}$ denotes the distance between the agent and the opponent.

• Reward for pushing the opponent further from the center of the ring proportional to $\exp(-d_{\text{center}})$, where $d_{\text{center}}$ denotes the distance of the opponent from the center of the ring.

We refer to [19] (Appendix D) for more RoboSumo details.

**MAgent Battle.**   Each agent receives an observation of a $13 \times 13 \times 9$ image with the following channels: its and opponent's team presence, its and opponent's team HP, its and opponent's team minimap, and its position [32]. The discrete action space has a dimension of 21 for moving around the gridworld and attacking the opponents. Lastly, reward is given as 5 for killing an opponent, -0.005 for every timestep cost, $0, 2$ for attacking an opponent, and -0.1 reward for dying. We refer to [32] for more MAgent details.

### G.2   Baseline Details

• LILI [8] maximizes the discounted return $v_{\theta^i}^i$ in the active Markov game:

$$\max_{\theta^i} v_{\theta^i}^i(s, \boldsymbol{\theta^{-i}}) := \max_{\theta^i} \mathbb{E}\Big[ \sum_{t=0}^{\infty} \gamma^t \mathcal{R}^i(s_t, \boldsymbol{a_t}) \Big|_{\substack{s_0 = s, \ \boldsymbol{\theta_0^{-i}} = \boldsymbol{\theta^{-i}}, \\ a_{0:T}^i \sim \pi(\cdot|s_{0:T}; \theta^i), \boldsymbol{a_{0:T}^{-i}} \sim \boldsymbol{\pi}(\cdot|s_{0:T}; \boldsymbol{\theta_{0:T}^{-i}}), \\ s_{t+1} \sim \mathcal{T}(\cdot|s_t, \boldsymbol{a_t}), \boldsymbol{\theta_{t+1}^{-i}} \sim \boldsymbol{\mathcal{U}}^{-i}(\cdot|\boldsymbol{\theta_t^{-i}}, \boldsymbol{\tau_t^{-i}})}} \Big]. \qquad (25)$$

We implement LILI by replacing the average reward target $y$ in Equation (12) with the discounted return target: $y = r^i + \gamma v_{\theta^i}^i(s', \hat{\boldsymbol{z}}^{-i\prime}; \bar{\psi}_\beta^i)$.

• MASAC [14] maximizes the discounted return $v_{\theta^i}^i$ in the stationary Markov game:

$$\max_{\theta^i} \rho_{\theta^i}^i(s, \boldsymbol{\theta^{-i}}) := \max_{\theta^i} \mathbb{E}\Big[ \sum_{t=0}^{\infty} \gamma^t \mathcal{R}^i(s_t, \boldsymbol{a_t}) \Big|_{\substack{s_0 = s, \\ a_{0:T}^i \sim \pi(\cdot|s_{0:T}; \theta^i), \boldsymbol{a_{0:T}^{-i}} \sim \boldsymbol{\pi}(\cdot|s_{0:T}; \boldsymbol{\theta^{-i}}), \\ s_{t+1} \sim \mathcal{T}(\cdot|s_t, \boldsymbol{a_t})}} \Big]. \qquad (26)$$

MASAC employs the framework of centralized training with decentralized execution [12] and has access to other agents' policies to perform optimization during training.

• **DRON [57]:** An approach that extends DQN [58] with opponent modeling by predicting both Q-values and current strategies of other agents. This baseline fails to predict future policies of others.

• **MOA [45]:** An approach that additionally optimizes the influence reward to consider influential actions to other agents. This baseline also has the discounted return objective.

## G.3 Hyperparameter Details

We use an internal cluster equipped with GPUs of RTX 3090 and CPUs of AMD Threadripper 3960X for choosing hyperparameters. We report the important hyperparameter values that we used for each of the methods in our experiments:

| Hyperparameter | Value |
|---|---|
| Critic learning rate $\alpha_q$ | 0.002 |
| Gain learning rate $\alpha_\rho$ | 0.02 |
| Actor learning rate $\alpha_\pi$ | 0.0005 |
| Inference learning rate $\alpha_\phi$ | 0.002 |
| Entropy weight $\alpha$ | 0.4 |
| Dimension of latent space $|z^{-i}|$ | 5 |
| Discount factor $\gamma$ | 0.99 |
| Batch size | 256 |

Table 2: IBS Experiment

| Hyperparameter | Value |
|---|---|
| Critic learning rate $\alpha_q$ | 0.0005 |
| Gain learning rate $\alpha_\rho$ | 0.02 |
| Actor learning rate $\alpha_\pi$ | 0.0001 |
| Inference learning rate $\alpha_\phi$ | 0.0005 |
| Entropy weight $\alpha$ | 0.3 |
| Dimension of latent space $|z^{-i}|$ | 5 |
| Discount factor $\gamma$ | 0.99 |
| Batch size | 64 |

Table 3: IC Experiment

| Hyperparameter | Value |
|---|---|
| Critic learning rate $\alpha_q$ | 0.01 |
| Gain learning rate $\alpha_\rho$ | 0.05 |
| Actor learning rate $\alpha_\pi$ | 0.001 |
| Inference learning rate $\alpha_\phi$ | 0.01 |
| Entropy weight $\alpha$ | 0.35 |
| Dimension of latent space $|z^{-i}|$ | 5 |
| Discount factor $\gamma$ | 0.99 |
| Batch size | 64 |

Table 4: IMP Experiment

| Hyperparameter | Value |
|---|---|
| Critic learning rate $\alpha_q$ | 0.0002 |
| Gain learning rate $\alpha_\rho$ | 0.2 |
| Actor learning rate $\alpha_\pi$ | 0.0001 |
| Inference learning rate $\alpha_\phi$ | 0.0002 |
| Entropy weight $\alpha$ | 0.01 |
| Dimension of latent space $|z^{-i}|$ | 10 |
| Discount factor $\gamma$ | 0.99 |
| Batch size | 256 |

Table 5: RoboSumo Experiment

| Hyperparameter | Value |
|---|---|
| Critic learning rate $\alpha_q$ | 0.001 |
| Gain learning rate $\alpha_\rho$ | 0.2 |
| Actor learning rate $\alpha_\pi$ | 0.0005 |
| Inference learning rate $\alpha_\phi$ | 0.001 |
| Entropy weight $\alpha$ | 0.01 |
| Dimension of latent space $|z^{-i}|$ | 10 |
| Discount factor $\gamma$ | 0.99 |
| Batch size | 256 |

Table 6: Battle Experiment

## H  Additional Evaluation

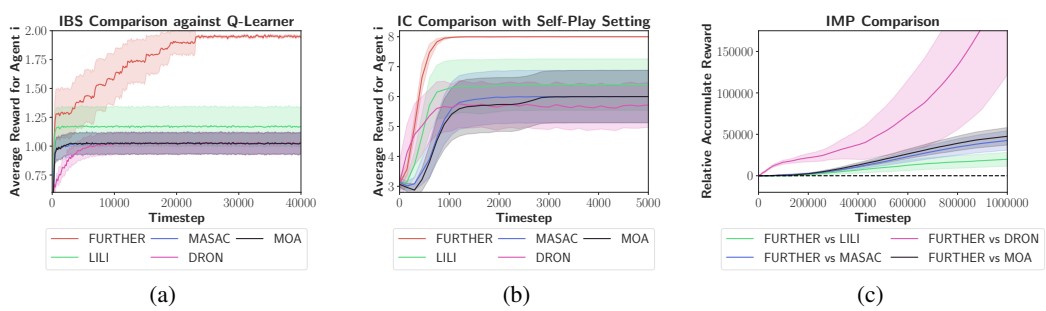

Figure 7: **(a)** Convergence in IBS. The FURTHER agent achieves convergence to its optimal pure strategy Nash equilibrium. **(b)** Convergence in IC with self-play. The FURTHER team shows better converged performance than baselines. **(c)** A competitive play in IMP between FURTHER and baseline methods. FURTHER receives higher rewards than baselines over time.

We show additional results about DRON and MOA in playing the iterated matrix games (see Figures 7a to 7c). Because DRON and MOA also suffer from myopic evaluation, we generally observe the sub-optimal performance of these baselines in our evaluations. In particular, DRON does not consider the underlying learning of other agents, resulting in the FURTHER agent easily exploiting the DRON opponent in Figure 7c. We also observe that, while MOA's optimization of the influence reward can effectively learn coordination in sequential social dilemma domains [49, 45], this influence reward optimization may not be useful in the competitive setting.

## I  Limitation and Societal Impact

FURTHER has a limitation that the framework does not consider an agent $i$'s own non-stationary policy. As discussed in Section 3, it is ideal to maximize the average reward over the space of joint update functions, including $i$'s own update function. However, it is computationally intractable to solve long horizon meta-learning by considering $i$'s own policy dynamics, and this remains an active area of research [9, 29, 30]. Instead, we take a practical approach by assuming $i$'s fixed stationary policy. Taking an agent's own non-stationary policy into account is one of the future directions. We also model the period as $k = 1$ for simplicity in our experiments, and studying how varying $k$ has a potential effect on performance is another future direction. Regarding the societal impact, while FURTHER can achieve a better social outcome in cooperative and self-play settings, a FURTHER agent aims to influence other agents to converge to desirable policies from its perspective. As such, there can be applications, where the framework may lead to negative societal impacts by taking advantage of other agents' defective decision-making.