# OpenReview forum: "Influencing Long-Term Behavior in Multiagent Reinforcement Learning"
_NeurIPS.cc/2022/Conference — NeurIPS 2022 Accept_

### Official Review · Reviewer_hBd2 · 2022-07-06

**Rating:** 6
**Confidence:** 4
**Soundness:** 3 good
**Presentation:** 3 good
**Contribution:** 3 good

**Summary:**

This paper considers the issue of non-stationarity in decentralized multi-agent reinforcement learning by developing a new framework, called active Markov game, which explicitly models changing parameters in the environment transition. Then, following the concept of active Markov game, a more general solution concept considering the average reward is discussed. This solution concept is further implemented within the SAC algorithm with variational inference adopted to model the changing policy parameter. Experiments are conducted over matrix games, a zero-sum MuJoCo battle environment and a grid-world mixed cooperative and competitive game.

**Questions:**

1. From Fig.1, we cannot see any multi-agent specific factors. So what if the number of agents degenerates to 1? Does an active Markov game degenerate into a stationary Markov game? I feel so. Some discussions will be appreciated.
2. In the second line of equation (3) (above line 109), there is a summation over all the states. Isn't a density term over states missing?
3. Again in equation (3), how do we choose $k$ in practice?
4. In line 169, the authors state that ``_while maximizing over the space of jointly-stable periodic distributions confines the search space of stationary periodic distributions, this still allows convergence to the **best possible active equilibria**._''. So why could the method converge to the *best* equilibrium? Also, isn't the equilibrium unique under the Jointly-Stable Periodic Distribution? What's the gap between the two?

**Strengths And Weaknesses:**

## Strength:
This paper proposes a novel framework with rigorous theoretical studies. I like the idea of introducing the long-term effect into the MARL framework without simply considering the stationary converged policies. The implementation of the method is also generic. The experiments are conducted over 3 domains, which is great, even from an empirical perspective.


## Weakness
The paper is generally great, but the clarity and evaluation can be possible further improved. Some of the major concerns are listed below. More detailed questions can be found in the next section.
1. **Formulation of active Markov game**. I feel the design of the active Markov game is a bit less motivated. I think there are multiple ways to incorporate the changing policy into the formulation but why is the active Markov game finally chosen? The most confusing part is that the current formulation of the active Markov game, at least from the diagram in Fig.1, looks like a meta-learning formulation, i.e., each player adapts within a single episode. Such an impression comes from a lot of definitions: for example, $\mathcal{U}^i$ (line 79) is defined over a single transition. However, from the content of the paper, it seems that the correct understanding is that the active Markov game intends to model the overall learning process. I think the authors could possibly improve the paper by better clarifying the insights of the active Markov game.
2. **Evaluation of competitive games**. In general, I do feel the evaluation results on the matrix games is sufficient. However, the authors use cross-play between policies from different methods as the evaluation metric on RoboSumo and MA-Battle, which looks less convincing to me. Frankly, I would feel more necessary to show the learned strategies (or converged equilibria) by different algorithms than to show the cross-play results. There are many factors influencing the ad hoc play performance of a policy and there are lots of straightforward techniques that can help improve the zero-shot generalization capabilities, such as population training, fictitious co-play, adversarial training, etc. The current results may suggest that the FURTHER policy is more robust but I don't feel the present results could sufficiently imply the conclusion that the FURTHER converges to _a better equilibrium_. So the paper can be stronger if the evaluation of complex games is more comprehensive.

## Minor Issues
Many equations are split over two lines, e.g., Eq (2) (5) (6). I would suggest the authors put the right-hand side of the equation into a single line.

---

> ### Author Response · Authors · 2022-08-02
> **Response to Reviewer hBd2 (Part 1/2)**
>
> We thank the reviewer for the valuable comments and positive evaluation. We have addressed individual concerns below and have carefully updated the main paper and appendix based on your helpful feedback.
>
> **Q1.** Formulation of active Markov game
> **A1.** Thank you for this insightful question. Regarding the formulation, we are unsure what alternative strategies could be possible for incorporating the changing policies on the formulation level. We would like to ask the reviewer whether some examples of alternatives to the active Markov game could be provided so that we can provide theoretical comparisons and clarify better in the final paper.
>
> We agree with the reviewer regarding the possible confusion with Figure 1. Your understanding is correct that we are in the continuing environment setting and that we are modeling and thus learning to influence the overall learning process. We have updated Figure 1 in an attempt to clarify this.
>
> **Q2.** Evaluation of competitive games
> **A2.** Thank you for raising this important question. We first note that our evaluation metric is inspired by competitive multiagent frameworks (e.g., [Bansal et al., 2018](https://arxiv.org/pdf/1710.03748.pdf); [Baker et al., 2020](https://arxiv.org/pdf/1909.07528.pdf)), where these works report rewards as agents directly play each other and are learning over time. The minor difference in our paper is that, because showing rewards can be noisy and thus hard to interpret especially in zero-sum games, we report the relative accumulated reward (line 275) for clarity. In addition, it is generally difficult to find possible equilibria in the complex domains of RoboSumo and MA-Battle, compared to the matrix games, so this metric was our natural choice. On the other hand, we also agree with the reviewer's insight that the results show that FURTHER achieves higher rewards over time, but they do not necessarily correspond to FURTHER converging to a better equilibrium in these complex domains. Based on your question, we will provide an analysis of the degree of convergence of the joint set of policies to a single cycle of a certain length in the final paper.
>
> **Q3.** Active Markov game (Figure 1)
> **A3.** Thank you for this interesting feedback. As detailed in Section 2.1, the bolded action (without the superscript) represents a joint action $a_{t}=(a^i_t,a^{-i}_{t})$. As such, the state transition in Figure 1 is a function of joint action, following the standard multiagent framework of the Markov game [44]. If the number of agents is 1 (i.e., $a_t=a^i_t$), the active Markov game does not degenerate into the stationary Markov game, but corresponds to a single agent meta-learning setting, where a meta-agent $i$ searches for its desirable update function $\mathcal{U}^i$ w.r.t. its non-stationarity policy dynamics [19].
>
> **Q4.** Missing density term over states for Equation (3)
> **A4.** We note that $\mu_k$ in the second line of Equation (3) is the density term, it should consider a subset of states ($\{s_1,...,s_k\}$) and policies $(\{\theta_1,..,\theta_k\})$ that satisfy Equation (2).
>
> **Q5.** Choosing the period $k$
> **A5** While we outline the very general theoretical solution concept of active equilibria w.r.t. a period $k$, we modeled the period as $k=1$ for simplicity in our experiments and developed a practical soft reinforcement learning approach based on that. Note that we can extend the current FURTHER implementation to settings with $k>1$ by sampling $k$ states and policies that are consecutive within each batch. We appreciate your question as it highlights an area that we should have emphasized more in our initial submission. As such, we will perform an ablation study varying $k$ in this way to report its potential effect on performance in the final paper.

---

> > ### Author Response · Authors · 2022-08-02
> > **Response to Reviewer hBd2 (Part 2/2)**
> >
> > **Q6.** Convergence to the best possible active equilibria
> > **A6.** Thank you for your note about the confusing comment we made in the submission about the best possible active equilibria. We have edited the description in lines 165-166 in order to clarify.
> >
> > As we highlight in Figure 2, all active equilibria are jointly-stable periodic distributions, but on the other hand, it is certainly not the case that all jointly-stable periodic distributions are active equilibria. A jointly-stable periodic distribution can be called an active equilibrium if Equation (4) in Definition 3 is satisfied. As such, we do not expect there to only be one active equilibrium, and there can be multiple. Our method computes gradients with respect to evaluations of the current jointly-stable periodic distribution and tries to find the best one it can with gradient descent. Unfortunately, it is still unclear how to formally guarantee convergence to the best possible active equilibrium for the particular deep RL approach we consider in our experiments, rather the joint policy space may converge to a jointly-stable periodic distribution that is only a local optimum in practice. It is also important to note as we highlighted in lines 115-118 that active equilibria can only be reached if all agents properly maximize for their own average reward, so it is possible that an agent that optimizes over the set of jointly-stable periodic distributions may find an even more beneficial solution for itself than any single active equilibrium by exploiting the sub-optimal update functions of other agents in the environment.
> >
> > **Q7.** Split over two lines
> > **A7.** Following your helpful suggestions, we have put the right-hand side of Equations (2), (5), and (6) into a single line.

---

> > > ### Author Response · Authors · 2022-08-08
> > > **We look forward to your valuable feedback**
> > >
> > > Thank you again for your positive review. The author-reviewer discussion period ends on August 9th, and we hope our response and revision addressed your concerns to support our paper for acceptance.

---

> > > ### Comment · Reviewer_hBd2 · 2022-08-09
> > > **Update**
> > >
> > > I appreciate the explanations. I remain positive about this paper.
> > >
> > > Regarding the suggestion on the definition, I think it is definitely okay to assume a continual RL setting. I would just suggest to put some discussions or remarks on the equivalence between standard episodic RL and the continual RL settings to make the presentation clearer.

---

> > > > ### Author Response · Authors · 2022-08-09
> > > > **Thank you for your response**
> > > >
> > > > We appreciate your positive evaluation of our paper. Yes, following your helpful suggestion, we will make the presentation clearer in the final paper by adding discussions on the equivalence between standard episodic and continual RL settings.

---

### Official Review · Reviewer_h5DN · 2022-07-09

**Rating:** 4
**Confidence:** 5
**Soundness:** 2 fair
**Presentation:** 3 good
**Contribution:** 2 fair

**Summary:**

The paper propose a multiagent learning framework by explicitly modeling the long-term behavior of other interacting agents in the environments and show good convergence performance by playing against different types of opponents or self-play.


**Questions:**

see pros and cons section

**Ethics Review Area:**

["I don’t know"]

**Limitations:**

see pros and cons section

**Strengths And Weaknesses:**

The paper provides a comprehensive and formal modeling and theoretical analysis of the mutiagent learning problem with opponent modeling and also gives a practical implementation version using VAE with good convergence performance.

One concern is the marginal contribution of the proposed framework. The idea of explicitly modeling the behavior of opponents is not new and it seems the main novelty of this work is to explicitly modeling the long-term behavior of opponents instead of myopic prediction. Another weakness is that the authors compared with two baselines which are not representative of the literature. There are quite a number of highly related works the authors may discuss but should compare with directly if possible, to name a few, [1-3]

[1] Achieving cooperation through deep multiagent reinforcement learning in sequential prisoner's dilemmas, DAI 2019
[2] Opponent Modeling in Deep Reinforcement Learning, 2016
[3] A Deep Bayesian Policy Reuse Approach Against Non-Stationary Agents. NeurIPS2020.

Another suggestion is to include the sequential PD benchmarks proposed in  AAMAS17 paper" Multi-agent Reinforcement Learning in
Sequential Social Dilemmas", which would strengthen the evaluation results. I would be willing to increase the score if the concerns can be addressed in the rebuttal phase.

---

> ### Author Response · Authors · 2022-08-02
> **Response to Reviewer h5DN**
>
> We appreciate your helpful comments about our paper. We have answered your concerns below and carefully incorporated your insightful feedback into a revision of our main paper and appendix.
>
> **Q1.** Main contribution of FURTHER
> **A1.** Our main contribution is our principled framework that enables each agent to directly account for the impact of its behavior on the limiting set of policies that other agents will converge to. However, it is difficult to achieve this farsighted evaluation in practice because increasing the number of anticipated updates to the limit is computationally intractable. Furthermore, setting the discount factor $\gamma\rightarrow 1$ results in unstable learning [15]. To achieve our important objective, we first closed the theoretical gap in the learning-aware MARL literature by defining a new theoretical solution concept of active equilibria as agents consider the limiting policies of other agents as time approaches infinity in Section 2. Based on our theoretical contributions, we developed a practical optimization objective that learns beneficial policies in the space of the jointly-stable periodic distributions and enables each agent to consider its impact on the limiting policies of other agents. In conclusion, FURTHER provides the first theoretical, principled, and practical framework for achieving the farsighted evaluation, which was not possible in related MARL frameworks.
>
> **Q2.** Baseline comparison
> **A2.** We carefully selected a set of baselines to allow us to understand the nature of the improved long-term performance of FURTHER and note that reviewers FNVy and hBd2 complimented our evaluation. In particular, the selected baselines are closely related to our technical approach, optimizing different objectives w.r.t. the policy update function $\mathcal{U}^{-i}$. Because LILI [8] and MASAC [14] optimize the discounted return objective with and without modeling $\mathcal{U}^{-i}$, respectively (please refer to Equations (25) and (26) in Appendix G.2), our baseline choices enable us to separately analyze the effect of FURTHER's novel average reward objective. We also note that LILI and MASAC are highly representative frameworks in the literature, since LILI received the best paper award at CoRL-20 and MASAC builds on top of the popular framework of MADDPG [12]. We have provided additional clarity on this point in the revision (please refer to Section 4).
>
> While we provided a sufficient set of baselines in the submitted draft, we agree with your helpful suggestion that our empirical results can be further strengthened by adding more baselines. In the revision, we have added the suggested representative opponent modeling baseline ([He et al., 2016](https://arxiv.org/pdf/1609.05559.pdf) [40]) and incentive MARL baseline ([Jaques et al., 2018](https://arxiv.org/pdf/1810.08647.pdf) [34]). The new results in Figure 4 show empirically that FURTHER still achieves the best performance. We also have added more discussion including the papers you recommended in Section 5.

---

> > ### Author Response · Authors · 2022-08-08
> > **We look forward to your valuable feedback**
> >
> > Thank you again for your review. The author-reviewer discussion period ends on August 9th, and we hope our response and revision (with the newly added baseline results) addressed your concerns to increase your score.

---

### Official Review · Reviewer_FNVy · 2022-07-11

**Rating:** 6
**Confidence:** 3
**Soundness:** 3 good
**Presentation:** 3 good
**Contribution:** 3 good

**Summary:**

This paper aims to address the non-stationarity coming from the active learning agents in multi-agent reinforcement learning. It proposes a novel framework for considering the convergence of policies by accounting for the influence on others. This paper also develops a new algorithm for optimizing agents under the proposed framework. Empirical results show desirable performance compared with SOTA.

**Questions:**

Please see Weaknesses 1, 3, 4.

**Limitations:**

I do not see any potential societal impact not discussed.

**Strengths And Weaknesses:**

**Strengths:**
This paper addresses one of the most important issues in MARL, namely the non-stationarity of the environment from the perspective of individual agents. The framework is novel from my understanding. The experimental environments are well selected.


**Weaknesses:**
1. FURTHER uses VAE to bypass the issue of unobservable policy parameters and updating dynamics. It is a convenient choice but not immediately clear to me how to go from proposition 3 (Active Average Reward Policy Gradient Theorem) to Equation (12) which is using the latent variable. It seems that a very complex policy gradient suddenly reduces to something extremely simple. If the latent variable represents the inferred policy parameters $\boldsymbol{\theta}_{t}^{-i}$, where is the updating dynamics $\mathcal{U}^{-i}$? I hope the authors could provide more explanation.
2. VAE is notoriously difficult to train and usually requires very delicate hyperparameter fine-tuning, even for stationary data like images. I suspect it is even more difficult for non-stationary distribution. If the quality of the learned latent space is bad (e.g. mode collapse), the algorithm could break down easily.
3. How is Equation (14) generalized to continuous action space? There is a continuous action space experiment (RoboSumo) but I did not see an algorithm for that.
4. Although there is some discussion about the relationships between the solution concepts in Section 2.3, I still do not quite understand the intuition of why searching for Active Equilibria would result in more optimal equilibrium as argued in the experiment analysis. If LILI suffers from myopic evaluation due to the discount factor $\gamma$, would it be helpful to set $\gamma \to 1$ in the episodic setting (assuming a finite horizon)?


**Summary:**
Overall I enjoy reading this paper as an attempt to solve a significant problem. However, I found it is hard to understand the whole picture given the gap and missing information as discussed in the weaknesses. I am willing to increase my score if the authors could address my concerns.

---

> ### Author Response · Authors · 2022-08-02
> **Response to Reviewer FNVy**
>
> We would like to thank the reviewer for providing such insightful comments on our work. We have addressed individual concerns below and have updated the main paper and appendix accordingly.
>
> **Q1.** Clarification on VAE and updating dynamics
> **A1.** Thank you for your important question. As Appendix E details, we set the period $k=1$ in Proposition 3 for simplicity and developed the corresponding practical soft reinforcement learning optimization in Equation (12) (we note that the current implementation can be extended to settings with $k>1$ by sampling $k$ states and policies that are consecutive within each batch). Importantly, the encoder in VAE represents the policy dynamics of other agents $\mathcal{U}^{-i}$ with parameters $\phi^i_\text{enc}$ and predicts the next latent strategies of other agents
>  $\hat{z}^{-i}_{t+1}\sim\hat{\mathcal{U}}^{-i}(\cdot|\hat{z}^{-i}_t,\tau^i_t;\phi^i_\text{enc})$.
>
> Then Equation (12) applies the bootstrapping technique [18] and uses the encoder to compute the target value $y$ by implicitly considering $\hat{z}^{-i}_{t+1},...,\hat{z}^{-i}_\infty$. We have updated the main text in Section 3.2 and Appendix F to clarify our notation and implementation.
>
> **Q2.** Generalization to continuous action space
> **A2.** Thank you for pointing out this issue. Your observation is correct that Equation (14) is for discrete action space only [30]. We use a different optimization for continuous action space [14, 26], and we have included this optimization in the revision (please refer to Equations (22) and (23) in Appendix E.2).
>
> **Q3.** Motivation for finding active equilibria
> **A3.** We define a general solution concept of active equilibria in Section 2.2, where no agents can further optimize its average reward as time approaches infinity. Intuitively, active equilibria provide a \textit{superset} of existing solution concepts (please refer to Figure 2), so searching for the best active equilibrium would result in as good or better long-term performance compared to finding the best solution among other concepts.
>
> Regarding LILI with a discount factor $\gamma\rightarrow 1$, this baseline method would perform similarly to FURTHER for episodic settings with a finite update horizon $T$. However, we note that a finite horizon $T$ cannot capture non-stationary policies at convergence (e.g., cyclic convergence [4]), so episodic settings do not fully represent the non-stationarity in MARL. Instead, we propose to consider continuous settings with $T \rightarrow \infty$ to fully represent the non-stationarity. In these continuous settings, it is well known that the discount factor limits the effective horizon to $1/(1-\gamma)$ [15]. Also, as our experiment in Figure 5a shows, $\gamma\rightarrow 1$ results in increasingly unstable learning in LILI. Meanwhile, FURTHER provides a principled framework to consider the limiting policies of other agents with an infinite horizon $T$.

---

> > ### Author Response · Authors · 2022-08-08
> > **We look forward to your valuable feedback**
> >
> > Thank you again for your review. The author-reviewer discussion period ends on August 9th, and we hope our response and revision addressed your concerns to increase your score.

---

> > > ### Comment · Reviewer_FNVy · 2022-08-08
> > > **Response to Authors**
> > >
> > > Thanks for the clarification. It addressed most of my concerns. I will increase my rating to "Weak Accept".

---

> > > > ### Author Response · Authors · 2022-08-09
> > > > **Thank you for your response**
> > > >
> > > > We are happy that our response addressed your concerns. We also appreciate raising your score and supporting our paper.

---

### Official Review · Reviewer_kSMF · 2022-07-11

**Rating:** 6
**Confidence:** 4
**Soundness:** 3 good
**Presentation:** 3 good
**Contribution:** 3 good

**Summary:**

The authors propose an interesting framework to consider the limiting policies of other agents as time approaches infinity. The average reward of each agent is maximized by developing a new optimization objective that directly considers the impact of its behavior on the set of limiting set policies that the other agents will converge on. Better long-term performance than the state-of-the-art baseline is demonstrated over a range of different multi-agent benchmark domains.

**Questions:**

* From my point of view, the perspective of this work is very similar to the concept of incentivize, but I don't see any discussion by the authors about incentivize related, so why not? Also why no comparison? For example.
    * Learning to Incentivize Other Learning Agents
    * Adaptive Incentive Design with Multi-Agent Meta-Gradient Reinforcement Learning


**Ethics Review Area:**

["I don’t know"]

**Strengths And Weaknesses:**

Strengths：
* Motivation of the paper was interesting and very important
* The writing was very clear and the drawings helped with understanding

Weaknesses：
* The comparison method in the experimental part seems to be relatively less
* No consideration of "incentivize" MARL-related work

---

> ### Author Response · Authors · 2022-08-02
> **Response to Reviewer kSMF**
>
> Thank you for your kind words about our paper and helpful feedback. We have addressed each comment individually below and incorporated your feedback into a revision of our main paper and appendix.
>
> **Q1.** Relatively less baseline comparison
> **A1.** During our submission, we attempted to carefully select baselines to fully understand the improved long-term performance of FURTHER and note that reviewers FNVy and hBd2 compliment our evaluation. In particular, the selected baselines are closely related to our technical approach, optimizing different objectives w.r.t. the policy update function **$\mathcal{U}^{-i}$**. Because LILI [8] and MASAC [14] optimize the discounted return objective with and without consideration of **$\mathcal{U}^{-i}$**, respectively (see Equations (25) and (26) in Appendix G.2), our baseline choices enable us to analyze the effect of FURTHER's novel average reward objective. We also note that LILI and MASAC represent state-of-the-art frameworks in the literature, where LILI received the best paper award in CoRL-20 and MASAC extends the seminal MADDPG framework [12] based on soft reinforcement learning. That said, we agree with your suggestion that our empirical results can be further strengthened by adding more baselines. In the revision, we have added two additional representative baseline frameworks based on incentive MARL [34] and opponent modeling [40], and empirically show that FURTHER still achieves the best long-term performance (please refer to the new results in Figure 4 and Questions 1-3 in Section 4).
>
> **Q2.** Related works on incentive MARL
> **A2.** Following your helpful feedback, we have added a discussion about related works on incentive MARL in the revision (please refer to Section 5). As noted above, we also have added an incentive MARL baseline in the revision (please refer to the new results in Figure 4 and Questions 1-3 in Section 4). We originally considered the suggested paper of [Yang et al., 2020](https://papers.nips.cc/paper/2020/file/ad7ed5d47b9baceb12045a929e7e2f66-Paper.pdf). However, in this work agents directly provide incentive rewards to other agents, making the approach not directly applicable to the decentralized settings we consider. Instead, we have considered MOA ([Jaques et al., 2018](https://arxiv.org/pdf/1810.08647.pdf) [34]), which adds the influence reward only to the agent itself, as a baseline. Because incentive MARL optimizes an objective with a discount factor and thus also suffers from myopic evaluation bias, Figure 4 in the revision shows empirically that MOA does not achieve better long-term performance than FURTHER.

---

> > ### Comment · Reviewer_kSMF · 2022-08-03
> > **Reply To The Authors**
> >
> > Dear Authors.
> >
> > Thank you for your detailed replies, the current ones solved my confusion.

---

### Meta-Review · Area_Chair_7GWo · 2022-08-26

**Recommendation:** Accept
**Confidence:** Less certain

**Metareview:**

Reviewers found the long-term influence problem and proposed solution interesting and novel. During the rebuttal, important additional baselines and clarifying comments were added, addressing the most important reviewer concerns. Although the scores look borderline for this paper, all reviewers who engaged with the authors during the rebuttal are in favor of acceptance, and I agree.

**Award:**

No

---

### Decision · Program_Chairs · 2022-09-14

Accept